# Localized and global representation of prior value, sensory evidence, and choice in male mouse cerebral cortex

Kotaro Ishizu [1], Shosuke Nishimoto [1,2], Yutaro Ueoka [1] & Akihiro Funamizu [1,2] ✉

Adaptive behavior requires integrating prior knowledge of action outcomes and sensory evidence for making decisions while maintaining prior knowledge for future actions. As outcome- and sensory-based decisions are often tested separately, it is unclear how these processes are integrated in the brain. In a tone frequency discrimination task with two sound durations and asymmetric reward blocks, we found that neurons in the medial prefrontal cortex of male mice represented the additive combination of prior reward expectations and choices. The sensory inputs and choices were selectively decoded from the auditory cortex irrespective of reward priors and the secondary motor cortex, respectively, suggesting localized computations of task variables are required within single trials. In contrast, all the recorded regions represented prior values that needed to be maintained across trials. We propose localized and global computations of task variables in different time scales in the cerebral cortex.

Optimal decision-making requires estimating the current context from sensory inputs and making choices based on expected outcomes[1–5]. The neural basis of sensory-based context estimation is often investigated as perceptual decision-making in which humans and animals update a belief of context by accumulating sensory evidence[6,7]. In contrast, outcome-based action selection is investigated as value-based decisions in which animals update the expected value of action, defined as action value, with recent choices and outcomes[8–11]. Although some theoretical and physiological studies integrate perceptual and value-based decision-making[12,13], it is unclear how these processes are represented and integrated in various brain areas to optimize behavior.

In theory, prior knowledge of the context probability and action values are used to compute the decision threshold of choices[1,14,15]. For simplicity, context probability was fixed but choice outcomes were varied in our task to investigate prior values[16]. In the framework of perceptual decision-making, sensory inputs are often integrated in time to form a belief about current context. Uncertainty in context

estimation reduces by accumulating sensory inputs. Based on the decision threshold and the belief of context, subjects made a choice. The choice outcome then determined the prior values in the next trial so that the values are maintained across trials. Our study investigates how the brain integrates the prior value and sensory input for a current choice while maintaining the values for future actions.

The cerebral cortex, including the medial prefrontal cortex (mPFC), the secondary motor cortex (M2), and the auditory cortex (AC), are involved in perceptual or value-based decisions. The AC modulates sensory-evoked activity with reward expectations, attention, and task engagement[17–20], although previous study finds that the non-sensory modulations in AC alone are not enough to drive behavior[5]. The mPFC neuron activity represents action values, rewards, or choices[21,22], and these neurons are causally involved in value-based decisions[23,24]. A recent study reported that the mPFC represents a value[13] that combines prior values and sensory confidence. M2 represents action values faster than other brain areas, including the striatum and mPFC, in rats[10]. Part of M2, the frontal orientation field (FOF), is

[1]Institute for Quantitative Biosciences, University of Tokyo, Laboratory of Neural Computation, 1-1-1 Yayoi, Bunkyo-ku, Tokyo 113-0032, Japan. [2]Department of Life Sciences, Graduate School of Arts and Sciences, University of Tokyo, 3-8-2, Komaba, Meguro-ku, Tokyo 153-8902, Japan. ✉e-mail: funamizu@iqb.u-tokyo.ac.jp

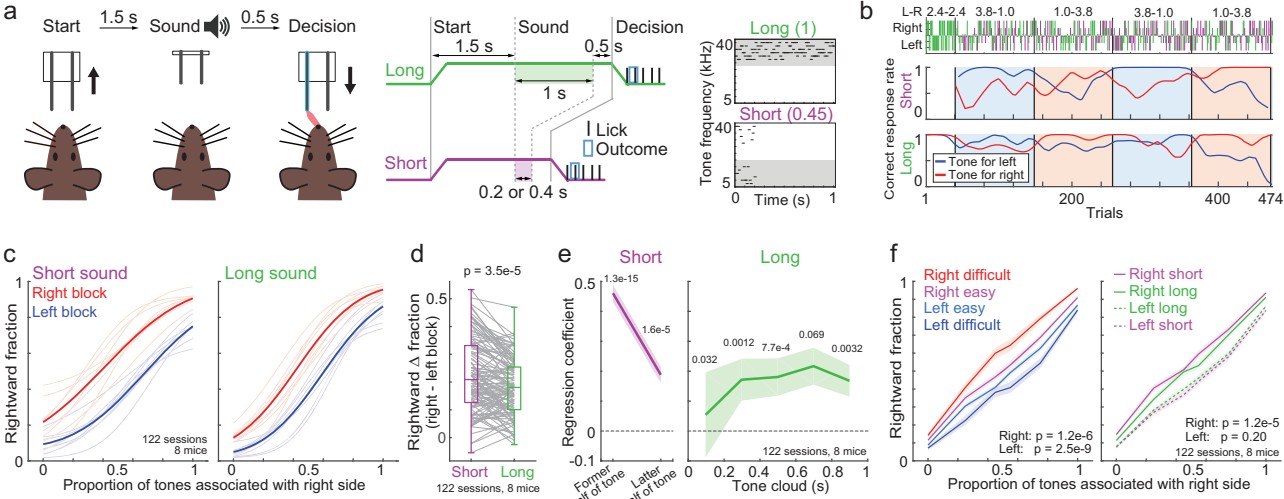

**Fig. 1 | Tone-frequency discrimination task with two sound durations and asymmetric-reward blocks. a** Task structure. Trial started by moving the spouts away from the mouse. After a long or short sound presentation, the spouts were moved and mice licked the left or right spout to receive a reward. Right panels show example tone clouds. This figure was published in iScience, 24/8, Akihiro Funamizu, Integration of sensory evidence and reward expectation in mouse perceptual decision-making task with various sensory uncertainties, 102826-0, Copyright Elsevier (2024). **b** Choice behavior in an example session. The correct-response rate is a moving average with a Gaussian function ($\sigma = 10$ trials). The top panel shows the stimulus length and proportion of tones associated with a left or right choice. Blue and red boxes show the left and right blocks. **c** Average psychometric function of choice behavior in all the 122 sessions from 8 mice. Overlayed traces show the average psychometric function in each mouse. **d** Stimulus duration affected reward-dependent choice biases. Rightward Δ fraction is the difference in the average fraction of rightward choices between the right and left blocks (two-sided

*t*-test in the linear mixed-effects model, 8 mice in 122 sessions) (central mark and edges in the box: median, 25th, and 75th percentiles; whiskers: most extreme data points not considered outliers (beyond 1.5 times the interquartile range)). **e** Psychophysical kernels. In short sound trials, logistic regression analyzed how the tone frequencies in former and latter half of sounds correlated to the choices. In long-sound trials, we divided the tone clouds with 5-time windows (Means and standard errors, two-sided *t*-test in the linear mixed-effects model, 8 mice in 122 sessions). **f** Sensory confidence affected choices in the next trial. Left: we categorized the rightward fraction of mice based on the tone difficulties in previous correct trials. The choices were compared between the previous easy and difficult stimuli. Right: The rightward fraction was categorized based on the tone durations in previous correct trials. The choices were compared between the previous short and long durations. Means and standard errors (two-sided *t*-test in the linear mixed-effects model, 8 mice in 122 sessions). Source data are provided as a Source Data file.

necessary in auditory perceptual decision-making tasks[25,26]. These studies suggest a hypothesis that the cerebral cortex, including the sensory and frontal cortices, are orchestrated to integrate prior knowledge and sensory stimuli. However, as previous studies mainly target specific brain areas in different tasks, it is unclear how the prior values and sensory inputs are represented and integrated in different cortical areas to optimize choices.

Here, we used a tone-frequency discrimination task for head-fixed mice based on previous studies[5,16,27], in which male mice discriminated a tone frequency to choose between a spout on the left and another on the right. Our task presented either a long or short auditory stimulus in each trial and biased the choice outcome according to a block of trials. We found that mice made accurate choices in response to long-duration tones compared to their choices in response to short-duration tones and made biased choices in the asymmetric-reward blocks, indicating that mice utilized both sensory uncertainties and reward expectations to guide behavior. The outcome experiences after short-duration or difficult-frequency sounds had large effects on the next trial compared to the effects of long-duration or easy-frequency sounds. Based on these results and a previous study[13], we modeled how the reward expectation of each choice (prior value) is sequentially modulated with sounds to compute action value within a single trial by reinforcement learning (RL). Our brain-wide electrophysiological study showed that the mPFC neurons gradually modulated the activity during sound and additively represented the prior values and choices[28], although we did not find brain regions representing the action values. The value computation requires sensory inputs that were decoded from the AC irrespective of asymmetric-reward blocks. The choices of the mice were represented by the M2 activity. These results suggest the localized computations of sensory inputs and

choices that were needed within single trials. In contrast, the prior values were equally represented by the activity across the three cortical regions, suggesting global maintenance. These results propose a localized and global computation of task variables that were needed within and across trials, respectively, in the cerebral cortex.

## Results

### Performance of mice depends on stimulus durations and asymmetric-reward blocks in a tone frequency discrimination task

To investigate how neurons in the cerebral cortex represent the prior value, sensory evidence, and choice, we updated our previous tone-frequency discrimination task (Fig. 1a)[16]. Mice were head-fixed and placed on a treadmill. Each trial started by moving the spout away from the mice. 1.5 s after moving the spout, we presented either a long (1.0 s) or a short sound. The duration of the short sound was either 0.2 or 0.4 s for 5 or 2 mice, respectively. In the other mouse, the duration of the short sound was 0.4 s for 1 session and 0.2 s for the remaining 4 sessions (8 mice in total). The sound consisted of short-duration pure tones (0.03 s), in which the frequency of each tone was either low (5–10 kHz) or high (20–40 kHz), named the tone cloud[5,27,29,30]. The proportion of high-frequency tones in the tone cloud was selected from 6 settings (0, 0.25, 0.45, 0.55, 0.75, and 1), and the dominant frequency determined the correct choice. The stimulus settings were named easy (0 or 1), moderate (0.25, 0.75), and difficult (0.45, 0.55) depending on the proportions. 0.5 s after the end of the sound, the spouts were moved forward, and the mice then chose the left or right spout. The correct or error choice provided 10% sucrose water (2.4 µl) or a noise burst (0.2 s), respectively.

After the first 40 trials in each session, we biased the reward size of the left and right spouts for correct choices in each block of

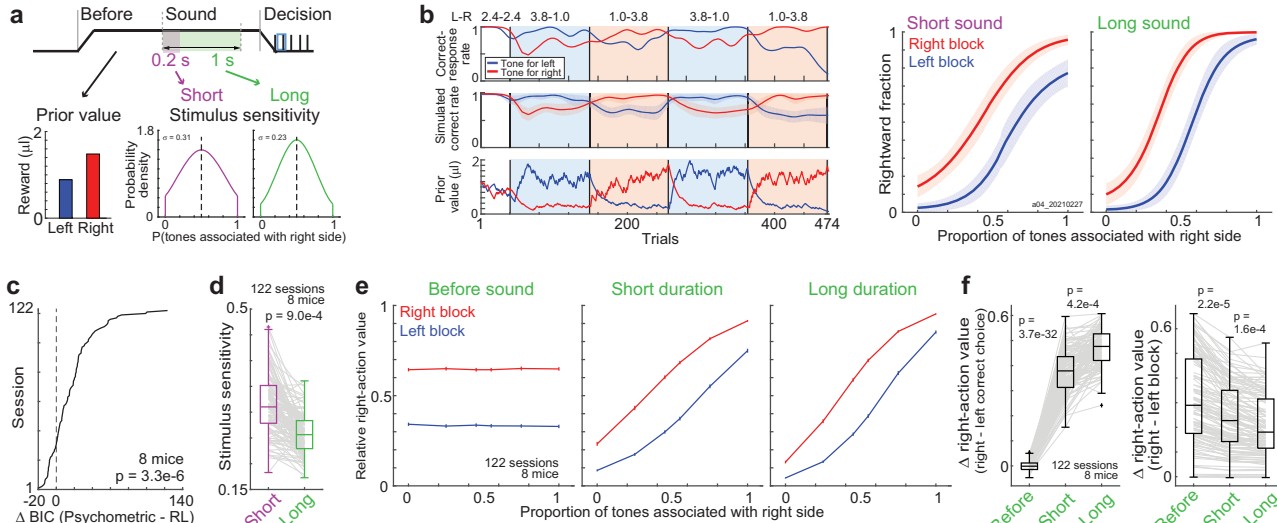

**Fig. 2 | Reinforcement learning model modulating the prior values with sensory inputs fit mouse behavior. a** Scheme of the reinforcement learning (RL) model. The RL model had reward expectations for left and right choice (prior values), which were shared between the long and short-sound trials. Stimulus sensitivities were independently set for the short and long sounds. **b** Choice simulation with the RL model in example session. Data are presented as in Fig. 1b. Based on the fitted parameters in RL model, we simulated the choices of a mouse 100 times. Means and standard deviations. **c** Comparison between the RL model and psychometric function. 122 sessions in 8 mice were sorted based on the difference of fitting between the models (Δ Bayesian information criterion: Δ BIC) (*p*-value in the two-sided *t*-test in linear mixed-effects model). **d** Comparison of stimulus sensitivities for long- and short-sound durations in the RL model (two-sided *t*-test in the linear mixed-effects model, 8 mice in 122 sessions) (central and edges of

the box: median, 25th, and 75th percentiles; whiskers: most extreme data points not considered outliers (beyond 1.5 times the interquartile range)). **e** Modulation of prior values with sensory inputs to compute action values. Relative right-action value was the ratio between the right and left action values which were plotted for before sounds, short-duration sounds, and long-duration sounds. Prior values were plotted before sound. Means and standard errors (8 mice in 122 sessions). **f** Summary of value modulation before and during sounds. Left: value differences between right- and left-correct trials. Right: value differences between right and left blocks (two-sided *t*-test in the linear mixed-effects model, 8 mice in 122 sessions) (central and edges of the box: median, 25th, and 75th percentiles; whiskers: most extreme data points not considered outliers (beyond 1.5 times the interquartile range)). Source data are provided as a Source Data file.

90–130 trials. The size of reward biases varied from 2.8–2.0 μl to 3.8–1.0 μl (large–small reward pair) for aligning the reward-dependent choice biases in each session. We defined the asymmetric-reward blocks as left or right blocks according to the large-reward side. Left and right blocks were alternated until the end of the session.

We analyzed the choice behaviors of 8 mice in 122 sessions. The high-frequency tones were associated with the correct choice in the left or right spout in 4 mice for each side. In the example session (Fig. 1b), the correct-response rates of the mouse were biased by the asymmetric-reward blocks. In short-sound trials, the choice behavior was less accurate and more biased toward the large-reward side than in long-sound trials. The block-dependent choice biases were larger in short-sound trials than in long-sound trials (8 mice in 122 sessions in Fig. 1c, d; each mouse in Supplementary Fig. 1a). Psychophysical kernels tested which tone-cloud timings and asymmetric-reward blocks correlated with choices (Fig. 1e, "Methods" section)[16,31–33]. In long-sound trials, the choices of mice were significantly correlated with almost the entire duration of sounds, suggesting that long sounds provided longer sound evidence than short sounds, thus resulting in accurate choices. The choices were also biased by the asymmetric-reward blocks (two-sided *t*-test in linear mixed-effects model, *p* = 1.3e-9 and 5.5e-8 for long- and short-sound trials, respectively; 8 mice in 122 sessions). Mice required 7 trials to switch the reward-dependent choice biases after the block changes (Supplementary Fig. 1b, c). These results were consistent with our previous study without neural recording[16].

A previous study showed that compared to high-confidence stimuli, low-confidence visual stimuli affected mouse choices in the next trial[13]. This matched our results showing that the previous rewarded trials with difficult tone clouds affected the choice in the next trial more than those with easy stimuli (Fig. 1f left), suggesting that sensory confidence affected choice behavior. This was further supported by

our results; the outcomes based on short sounds affected the choices more than those based on long sounds (Fig. 1f right).

## Reinforcement learning showed gradual modulation of prior values with sensory inputs to compute action values within single trials

To understand the computation behind the behavior, we used the RL model to analyze the choices the mice made (Fig. 2a). Based on the behavioral results (Fig. 1f) and previous studies[13,16,34], we hypothesized that the updating of reward expectations for left and right choices (i.e., prior values) depended on the tone frequencies and durations, i.e., sensory confidence about the current context. The RL model had independent perceptual uncertainties for the long and short sound durations. First, the prior values were used to compute the decision threshold. Based on the threshold and perceived sensory inputs, the model estimated the choice probability in each trial ("Methods" section). The action values were analyzed by multiplying the prior values with a contextual belief based on sensory inputs and used to update the prior values. Thus, the RL model made choices by combining the prior values and sensory inputs, while it updated the prior values using action values.

In the example session, our model captured the block-dependent changes in the correct-response rate of the mouse by updating the prior values (Fig. 2b left). The simulated choices captured the differences in choice accuracy and block-dependent choice biases between sound durations (Fig. 2b right). Our RL model matched the mouse choices better than the conventional psychometric functions or an RL model using only prior values (Fig. 2c and Supplementary Fig. 1d–g)[16]. Parameters in the RL model suggested that the stimulus sensitivity (i.e., perceptual uncertainty) for short sounds was significantly larger than that for long sounds (Fig. 2d).

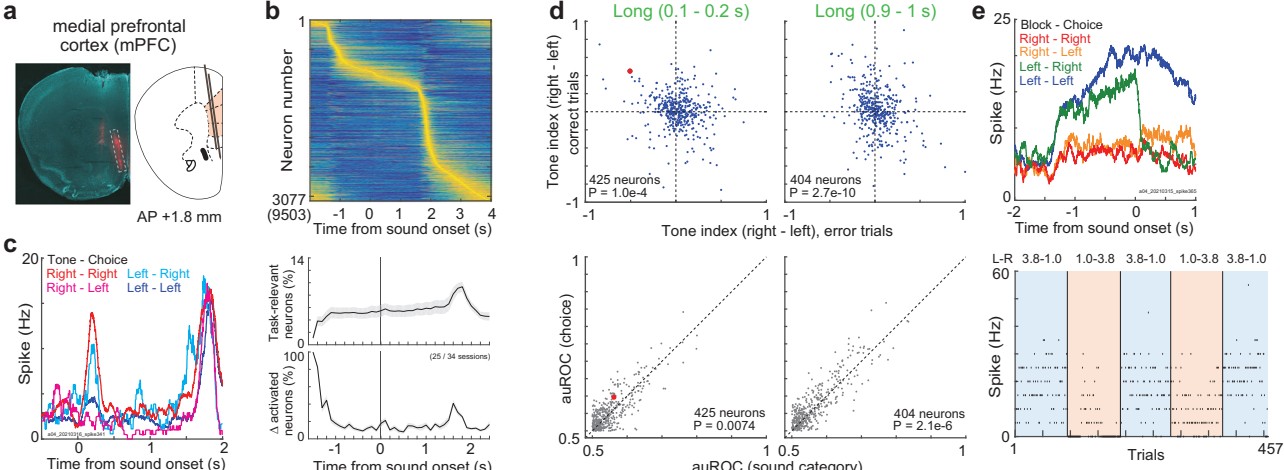

**Fig. 3 | Activity of neurons in the medial prefrontal cortex (mPFC). a** Location of Neuropixels 1.0 probe for the mPFC in an example mouse. The weak red color at the relative center of the brain slice was the location of the M2 probe. All the data in Fig. 3 were from long-sound trials. **b** Average activity of task-relevant neurons in the mPFC (3077 out of 9503 neurons). Top: average activity of each neuron was normalized between 0 and 1, and sorted by the peak activity timings. Bottom: proportion of task-relevant neurons in each time window of 0.1 s. Δ activated neurons was the proportion of task-relevant neurons newly recruited from the previous time window. Means and standard errors. **c** Average activity of one neuron after long-sound onset. The neuron showed increased activity when the mouse selected the right spout irrespective of sounds. **d** Tone indices at the beginning and end of

long sounds (top). Tone indices in correct and error trials were flipped, suggesting the choice representation. *P*-value in the inset shows the significance of the regression coefficient. Task-relevant neurons with a significant increase in activity at the analyzed time window were shown. Comparison of sound and choice decoding performance (two-sided Wilcoxon signed rank test) (bottom). The area under the receiver operating curve (auROC) evaluated the decoding performance ranging between 0.5 and 1. Red dot shows the example neuron in (**c**). **e** Average activity of one neuron before sound onset. The neuron showed increased activity in the left block irrespective of choices. Bottom: neural activity between −0.2 and 0 s from sound onset in each trial. Source data are provided as a Source Data file.

As our model parameterized the stimulus sensitivities for the duration of short- and long sounds, we could simulate how the prior values were gradually modulated by sensory inputs to form the action values before, during (short-sound duration), and at the end of long-sound presentations by showing the proportion of right-action values relative to the left (Fig. 2e). We found that the difference in the relative right-action values (Δ right-action value) between the right- and left-correct trials gradually increased during the sound presentations, while the difference between the left and right blocks decreased (Fig. 2f).

## The mPFC neurons additively represent the prior values and choices

During the tone-frequency discrimination task, we recorded the electrophysiological activity of neurons in the cerebral cortex with a Neuropixels 1.0 probe[35]. In each session, i.e., each day, we targeted one or two cortical areas and inserted Neuropixels probes. We repeated the procedure on multiple days to target the mPFC, M2, AC, or posterior parietal cortex (PPC). Automated spike sorting with Kilosort 3 showed that only less than half of neurons on average were detected from the PPC (55 neurons per session, 32 sessions) compared to the other 3 regions (mPFC, 280 neurons; M2, 113 neurons; AC, 207 neurons, two-sided Mann–Whitney *U*-test, $p = 2.3e-9 – 1.4e-4$) (see also "Methods" section). Thus, we decided not to analyze the data from PPC.

In the mPFC, we analyzed the neural activity of both hemispheres in 8 mice in 34 sessions (Fig. 3a and Supplementary Fig. 2). After spike sorting with Kilosort 3, we identified 9503 neurons in total based on the approximate depth of electrodes from the brain surface ("Methods" section). We then identified 3077 task-relevant neurons that significantly increased the activity at one of the 132 time windows during the task compared to the baseline activity ($p < 1e-10$ in the one-sided Wilcoxon signed rank test) (Fig. 3b top and Supplementary Fig. 3a–c). Each time window was 0.1 s between −1.5 and 2.5 s from sound onset in long-sound trials (40 windows) and between −1.5 and 1.7 s in short-sound trials (32 windows). The time windows were also set between −0.5 and 2.5 s from the choice timing both in long- and short-sound trials ($30 × 2 = 60$

windows; $40 + 32 + 60 = 132$ windows in total). The baseline for sound onset activity was defined as the activity at −0.2–0 s from the trial start, i.e., spout removal. The baseline for choice activity was defined as the activity at −0.2–0 s from the spout approaching, i.e., between the sound end and the choice. We focused on the neurons with increasing activity as the uniform criteria across regions, because previous study shows that the neurons with decreasing activity were less modulated by task variables than the increasing neurons in the mPFC[36]. In other areas, the increasing neurons were selectively analyzed to investigate the sensory representations or task-related activity[37,38]. We verified that the number of neurons significantly decreased the activity during sounds in the mPFC was at least 10 times smaller than the increasing task-relevant neurons (Supplementary Fig. 4).

We first found that, in addition to observing the ramping activity of neurons toward the end of sounds, the task-relevant neurons were recruited before and after sound onset of the long-sound trials (Supplementary Fig. 3d). In the mPFC, 5.6% of neurons on average were identified as task-relevant in each time window of 0.1 s (Fig. 3b bottom). 18.3% of neurons on average were newly recruited every 0.1 s during the task.

We analyzed the activity of each neuron in detail. Example neuron increased the activity when the mouse selected the right spout irrespective of tone categories, suggesting choice encoding (Fig. 3c). We investigated whether the mPFC neurons represented the choices or sounds by comparing the tone indices between the correct and error trials. The tone index was defined as the activity difference between the right- and left-category tones ranging between −1 and 1 ("Methods" section). If a neuron represented a tone category, the signs of tone indices between the correct and error trials became identical. However, the tone indices flipped for the choice representation. We found that the tone indices of mPFC neurons were flipped between the correct and error trials (Fig. 3d top). We also compared the decoding ability of sound and choice categories by analyzing the area under the receiver operating curve (auROC) ranging between 0.5 and 1[39]. We found that the choice decoding of single neurons was better than the

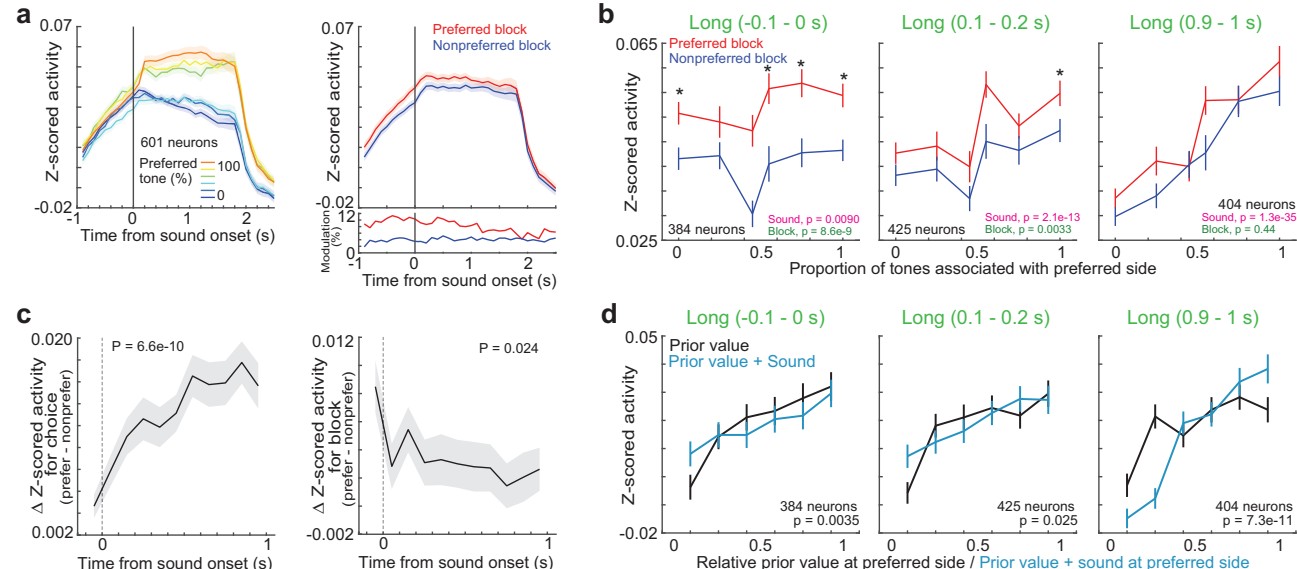

**Fig. 4 | Neurons in the medial prefrontal cortex (mPFC) additively represent the prior values and choices during sound. a** Means and standard errors of activity of task-relevant neurons. A subset of task-relevant neurons which increased the activity between −0.1 and 1.0 s were shown. The activity of neurons were aligned to their preferred tones (left) and their preferred blocks (right). The inset in bottom right shows the proportion of neurons significantly increased the activity at preferred (red) or nonpreferred (blue) blocks. The block-modulated neurons were defined with two-way ANOVA ($p < 0.01$). All the data in Fig. 4 were from long-sound trials. **b** Activity of task-relevant neurons before, at the beginning, and at the end of long sounds in correct trials. The activity was compared between sound categories and blocks with the two-way ANOVA ($p$-values in the bottom right). Medians and robust standard errors (*$p < 0.01$ in the two-sided Wilcoxon signed rank test; −0.1–0 s: $p = 0.0060, 0.026, 0.010, 3.0e-4, 9.0e-4, 7.6e-5$; 0.1–0.2 s: $p = 0.043, 0.015, 0.31, 0.025, 0.14, 7.6e-4$; 0.9–1 s: $p = 0.055, 0.054, 0.035, 0.025, 0.42, 0.26$). **c** Modulation

of choice- and block-dependent activity before and during sounds. The left panel shows the comparison of the activity between the preferred and nonpreferred sides. The right panel shows the comparison of the activity between the preferred and nonpreferred blocks. Medians and robust standard errors. Regression coefficient in the robust linear regression showed that the differences in activity between choices or blocks were significantly increased or decreased, respectively (two-sided $t$-test). **d** Activity of task-relevant neurons in different prior values or additive values (prior values+sounds). Medians and robust standard errors. We analyzed the Spearman partial correlations between the neural activity and prior values, or between the activity and additive values without the effect of running speeds of mouse. Two-sided Wilcoxon signed rank test compared the correlation coefficients for the prior values and additive values ($p$-value in each inset). The prior and additive values were correlated to the neural activity before and end of sounds, respectively. Source data are provided as a Source Data file.

sound decoding (Fig. 3d bottom). The absolute values of tone indices were increased from the initial to the end of long sounds in correct trials (two-sided Wilcoxon signed rank test, $p = 6.2e-6$, 425 neurons), suggesting that the choice representations became precise during sounds. The tone indices at the end of short sounds were correlated to those at the corresponding duration of long sounds, rather than those at the end of long sounds (Supplementary Fig. 5a), suggesting that the activity was gradually shaped from the sound onsets.

We also analyzed the neural activity before sound onset. Example neuron showed increased activity in the left block irrespective of choice (Fig. 3e). Before sound, the activity of mPFC neurons was modulated by blocks rather than by choices (Supplementary Fig. 5b). To analyze the left- and right-choice encoding neurons, and the left- and right-block encoding neurons at the same time, we aligned the activity of neurons to the preferred side. The preferred side (left or right choice) was defined once based on the activity at the time window with the largest difference from the baseline activity in long-sound correct trials (one-sided Wilcoxon signed rank test). For example, when the activity was higher in left-choice trials than in right-choice trials, the preferred side was defined as left. Then, the preferred block and tone were defined based on the preferred side: the preferred block had a large-reward amount on the preferred side, and the preferred tone was associated with the reward on the preferred side. For example, the preferred block and tone for a left-choice preferred neuron were defined as the left block and left-rewarded tone frequency, respectively. The aligned population activity of mPFC neurons gradually separated between tone categories in correct long-sound trials, while the block modulations were gradually decreased during trials (Fig. 4a).

We investigated how the task-relevant neurons in the mPFC changed their activity before and during the correct long-sound trials (Fig. 4b and Supplementary Fig. 5c). In these analyses, we analyzed the neurons that increased the activity in each time window. Before sound onset, the activity was significantly mainly modulated by blocks (two-way ANOVA). At the initial phase of sounds, activity was modulated both by the choices and blocks, while at the end of long sounds, activity was only modulated by choices. The neurons activated before sound in this analysis had almost no overlapping from the neurons that had sustained activity after the outcomes in previous trials (Supplementary Fig. 6)[40]. We then divided the task-relevant neurons into two groups (Supplementary Fig. 7): the former group increased the activity already before sound onset, while the latter group increased the activity only after sound presentations. We found that the former group showed gradual shifts of representations from block to choice modulations. In contrast, the latter group mainly showed choice modulation. We summarized how the block- and choice-dependent activity changed before and during long sounds (Fig. 4c). The choice and block modulations were increased and decreased, respectively, consistent with the gradual modulation of prior values with sensory inputs in the RL model (Figs. 2e, f).

To further investigate the relationship between the mPFC neurons and the values in RL model, we investigated the Spearman partial correlations between the neural activity and the prior values, the additive values (i.e., prior values+sounds) (Fig. 4d)[28], or the action values (Supplementary Fig. 8a) with eliminating the effects of running speeds of mice. Before sound, the mPFC activity was correlated to the prior values than the additive values (comparison of correlation coefficients among 384 neurons, two-sided Wilcoxon signed rank test

$p = 0.0035$), while at the end of sounds, the activity was correlated to the additive values ($p = 7.3e-11$) (Fig. 4d). Similar results were observed for the action values (Supplementary Fig. 8a). The correlations for additive and action values did not have significant differences at the end of sounds ($p = 0.052$). Linear regression with log-scaled neural activity showed that the mPFC neurons encoded the prior values before sound, while the neurons encoded both the choices and prior values during sound (Supplementary Fig. 8b, "Methods" section). These results suggest that the mPFC neurons additively represent the prior values and choices during sound.

We analyzed the neural activity of both hemispheres of M2 from 8 mice in 34 sessions. Similar to the results in the mPFC, M2 neurons represented the choices or asymmetric-reward blocks (Supplementary Figs. 9 and 8b). During correct long-sound trials, the choice modulation of activity gradually increased (Supplementary Fig. 9h). In contrast, the block modulation of activity was weak in M2 and did not show a decrease in block modulation during sounds. We then directly compared the choice and block modulations between the M2 and mPFC (Supplementary Fig. 10). First, the choice modulations were larger in the M2 than in mPFC, suggesting the choice representations in the M2. The prior modulations were larger in the mPFC, but the difference mainly came from the prior modulations before sound. These results suggest that the mPFC and M2 had similar activity patterns, but the mPFC and M2 had strong modulations on prior values and choices, respectively.

### Neuronal activity in the AC represents the prior values and sensory inputs

We analyzed the neuronal activity in both hemispheres of the AC from 7 mice in 29 sessions (Fig. 5a). We identified 2722 task-relevant neurons from the recorded 5994 neurons (Fig. 5b). At the sound onset, 70.7% of neurons on average were newly recruited (Fig. 5b bottom): the proportion of neurons was significantly larger than that in the mPFC of 17.7% (two-sided Mann–Whitney $U$-test, $p = 3.4e-7$, 16 and 25 sessions in the AC and mPFC). Example neuron in the AC showed increased activity during the right-rewarded tones irrespective of choices (Fig. 5c). In the population of neurons, the tone indices for correct and error trials had the same signs, suggesting sound encoding (Fig. 5d). The decoding performance of sound was better than choice in the AC neurons, although the neural activity was also modulated by choice during sound (Supplementary Fig. 11a, b). Consistent with the results in the mPFC, the tone indices at the end of short sounds were correlated with the tone indices at the corresponding duration of long sounds, rather than those at the end of long sounds (Supplementary Fig. 5a).

We investigated the activity of the AC before sound onset. Example neuron showed increased activity during the left block irrespective of choices, suggesting block encoding (Fig. 5e). We summarized the activity before and during the long-sound correct trials (Fig. 5f, g). 12.8% of neurons showed block-modulated activity before sound onset (Supplementary Fig. 5c), consistent with a previous finding of reward modulations in the AC[5,17]. We also confirmed that the activity of auditory cortical neurons before sound was modulated by the prior value rather than the tone category in the previous trial (Supplementary Fig. 11c). The block modulations were observed in both directions: activity in the preferred block was increased or decreased with respect to the preferred tone frequencies, resulting that the population activity was significantly increased at the preferred blocks than nonpreferred blocks in only one tone frequency (Fig. 5g left). Different from the mPFC and M2 (Fig. 4 and Supplementary Fig. 9), the sound-dependent activity of AC neurons rapidly changed at sound onset, while block-dependent activity was stable (Fig. 5h). Consistent with the observations of prior-modulated neurons in the AC (Supplementary Fig. 5c), linear regression analysis showed that the activity of AC neurons was modulated by prior values and sounds before and during sound (Supplementary Fig. 8b).

### Localized representation of sound and choice and global representation of prior values in the cerebral cortex

To further investigate the neural encoding of each cortical region and to compare the encoding across regions, we modeled the neural activity with a generalized linear model (GLM) with a software package of glmnet[41] (Fig. 6a, "Methods" section). The neural activity was averaged every 0.1 s and smoothed with a Gaussian filter ($\sigma = 0.25$ s). Our GLM had an equal number of kernels for sound categories, choices, and prior values, as the prior values were binarized depending on the dominant side. The prior value was estimated from the RL model, which achieved the maximum likelihood in each session (Fig. 2). Our GLM estimated the regression coefficients using all the data, except that the elastic net regularization $\lambda$ was determined with 10-fold cross-validation. We plotted how the GLM fit the example neural activity in the AC (single trials: Fig. 6a; average across trials, Fig. 6b). The full GLM succeeded in modeling that the neuron increased activity in response to left-rewarded tones irrespective of choices, while the GLM without sound kernels failed to model the sound representation.

By comparing the log-likelihood between the full GLM and GLM without sound, choice, or prior kernels, we analyzed the proportion of neurons encoding each task variable ($p < 0.01$ in the likelihood ratio test) (Fig. 6c). In this analysis, we analyzed the subpopulation of task-relevant neurons that significantly increased the activity at least once from the start of the trial to before the choice was made ($p < 1e-10$ in the one-sided Wilcoxon signed rank test). We found that a larger proportion of AC neurons represented sounds than the M2 and mPFC (chi-square test, $p = 1.1e-17$ and $3.5e-17$). M2 neurons represented choices more than the mPFC ($p = 2.5e-4$). The proportion of neurons representing prior values did not have significant differences across the three cortical regions ($p = 0.034 - 0.27$ before Bonferroni correction). We also compared the GLM fit across regions (Fig. 6d). The sound and choice were encoded in the AC and M2, respectively. The prior value was encoded in the mPFC and M2 rather than in the AC. The differences in prior encoding may come from the nonsignificant encoding neurons, as the proportion of prior-encoding neurons was similar across the three regions (Fig. 6c). Thus, the relative proportion of neurons encoding for sounds and choices differed across the 3 recorded regions, while there were no significant differences in the encoding of prior values among the regions.

We also investigated the proportion of neural encoding in each mouse (Supplementary Fig. 12a) and the proportion based on all the recorded neurons (Supplementary Fig. 12b). As the proportion of task-relevant neurons among all the neurons was small in the mPFC, the proportion of prior encoding neurons in the mPFC became small. One possible concern about the global representation of prior values was that the mouse body movements were correlated with the prior values and thus detected as the prior-value neural representation rather than the movement representation[42], although the running speeds of mice were already included in our GLM analysis. Among the 97 sessions, 8 mice with neural recording, we captured the facial motion of mice in 55 sessions, 4 mice by a camera (Supplementary Fig. 12c, d). We verified that the prior values and the facial motions were not significantly correlated (Supplementary Fig. 12e, f). We then found that the running speeds and prior values were significantly correlated in 55 out of 95 sessions for the AC, mPFC, or M2 recording, with 2 sessions with no recording of running speeds (Supplementary Fig. 12g, h). However, even with the uncorrelated 40 sessions, the neural encoding of prior values was observed (Supplementary Fig. 12i), supporting the global representation of prior values in the cerebral cortex.

### Localized readout of sound and choice and global readout of prior value from the AC, M2, and mPFC

It is important to investigate not only what neurons represent (encoding) but also what information other brain areas can extract from the brain region (i.e., decoding)[5,43]. We used sparse logistic

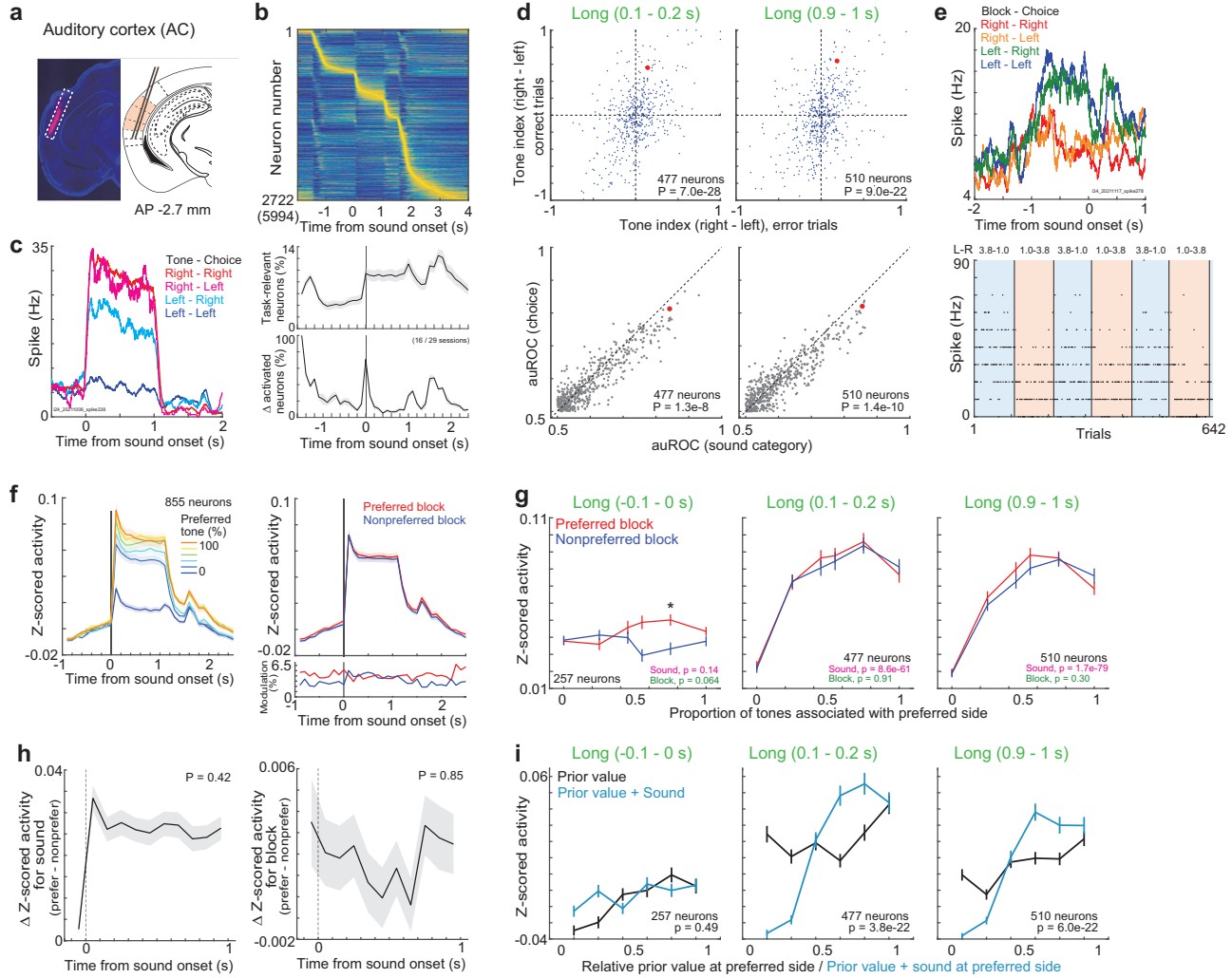

**Fig. 5 | Block and sound representations in the auditory cortex (AC). a** Probe location for the AC in an example mouse. **b** Average activity of task-relevant neurons in the AC. Data plots are same as in Figs. 3 and 4. Top: average activity of each neuron was normalized between 0 and 1, and sorted by the peak activity timings. Bottom: Means and standard errors. **c** Average activity of an example neuron during long sounds. The neuron showed increased activity when the tones associated with right reward were presented irrespective of choices. **d** Tone indices and area under the receiver operating curve (auROC) in task-relevant neurons. The tone indices of correct and error trials had same signs. *P*-value in inset shows the significance of regression coefficient (top). The discrimination of sound was better than that of choice, suggesting the sound representation (bottom: two-sided Wilcoxon signed rank test). **e** Average activity of an example neuron before sound onset. The neuron showed increased activity in left block irrespective of choices. **f** Means and standard errors of activity in task-relevant neurons. **g** Activity of task-

relevant neurons before, at the beginning, and at the end of long sounds in correct trials (*p*-value in each inset: two-way ANOVA). Population activity of AC neurons represented the sound frequencies. Medians and robust standard errors (*$p < 0.01$ in the two-sided Wilcoxon signed rank test; −0.1–0 s: $p = 0.18$, 0.40, 0.61, 0.032, 0.0073, 0.81; 0.1–0.2 s: $p = 0.18$, 0.94, 0.82, 0.73, 0.30, 0.24; 0.9–1 s: $p = 0.014$, 0.21, 0.13, 0.15, 0.99, 0.85). **h** Sound- and block-dependent activity before and during sounds. Difference in activity between preferred and nonpreferred tones (left), and between preferred and nonpreferred blocks (right) are shown. Medians and robust standard errors (*p*-values in the two-sided *t*-test in robust linear regression). **i** Activity of task-relevant neurons in different prior values or additive values (prior values+sounds). Medians and robust standard errors. Two-sided Wilcoxon signed rank test compared the correlation coefficients for the prior values and additive values (*p*-value in each inset). Source data are provided as a Source Data file.

regression (SLR) and analyzed the decoding performance of sound categories, choices, and binarized prior values (Fig. 7a). The three task variables were independently decoded in single trials with 10-fold CV. The CVs were repeated 100 times to reduce the noise from randomization. We first compared the decoding performance of sounds, choices, and prior values within each cortical region (correct rate: Fig. 7b; d prime: Supplementary Fig. 13a). In the decoding analyses, we used the sessions with simultaneously recorded activity of more than 20 neurons. Before sound onset, the prior readout was better than the sound and choice in all 3 regions (two-sided Wilcoxon signed rank test, $p = 5.4e\text{-}7\text{-}3.8e\text{-}6$). During long sounds, the AC had a better readout of sound than choice and prior values ($p = 8.1e\text{-}6\text{-}0.0017$). In M2, choice readout gradually improved during sounds (comparison between the

choice and prior decoding at initial and late sounds: $p = 0.13$ and $7.9e\text{-}6$), while the mPFC did not have significant differences between the choice and prior decoding even at the late phase of sounds ($p = 0.35$). In all 3 regions, the decoding performance of the prior value was above the chance level before and during long sounds (two-sided Wilcoxon signed rank test, $p = 5.4e\text{-}7\text{-}4.9e\text{-}4$), suggesting a stable prior readout. The prior value decoding at the block changes showed that the updating speed of value decoding did not have significant differences across the 3 regions (Supplementary Fig. 14a, b).

We next compared the decoding performance between the long- and short-sound trials (Fig. 7c). The choice decoding in both the M2 and mPFC was better in the long- than in short-sound trials, suggesting that these two regions were not simply decoded choice, but the

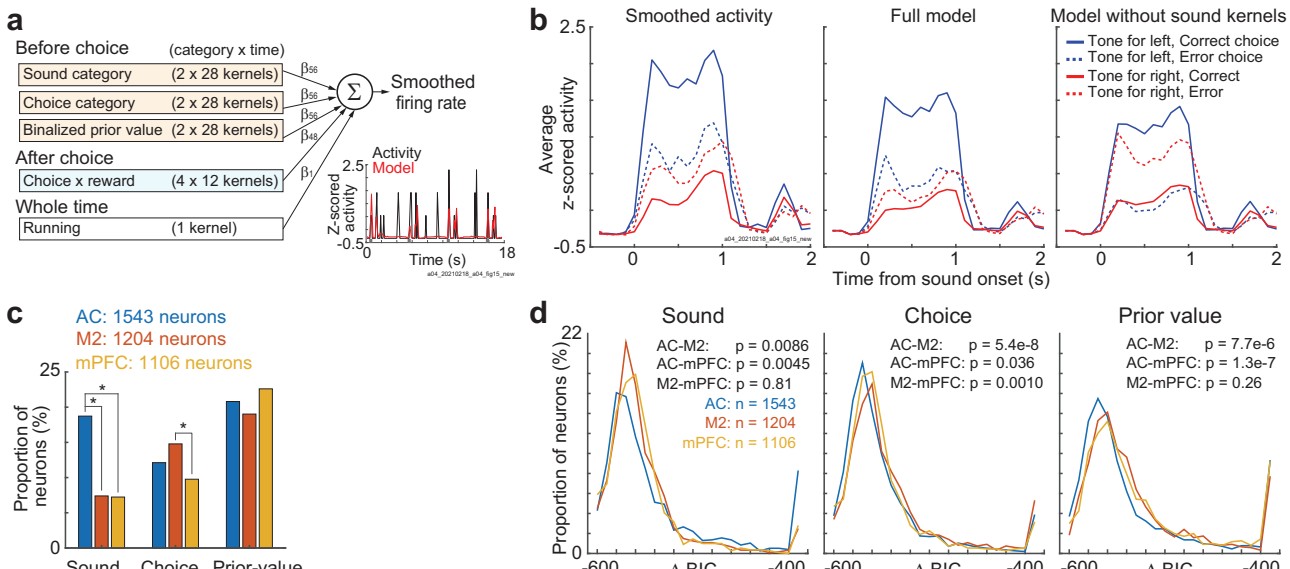

**Fig. 6 | Sound, choice, and prior value encoding in the auditory cortex (AC), secondary motor cortex (M2), and medial prefrontal cortex (mPFC).**
**a** Schematic of the generalized linear model (GLM) to test the neural encoding of task variables. Our GLM estimated the log-scaled smoothed activity of each neuron from the 217 kernels and constant term. The right bottom plot shows the smoothed activity and the output of the GLM in example trials in one auditory cortical neuron. **b** GLM in the auditory cortical neuron in (**a**). Left: average neural activity. The neuron showed increased activity during left-reward-associated tones. Middle: average trace of the full GLM. Right: average trace of the GLM without sound kernels. The GLM without sound kernels failed to model the sound encoding of the neuron. **c** Proportion of neurons significantly represented sound, choice, and prior value in each cortical region. We compared the log-likelihood between the full GLM and the model without corresponding kernels ($p < 0.01$ in the two-sided likelihood ratio test). The proportion of neurons were compared across regions with the two-sided chi-square test (*$p < 0.01$; sound, AC-M2, AC-mPFC, M2-mPFC, $p = 1.1e$-$17$, $3.5e$-$17$, $0.88$; choice, $p = 0.041$, $0.58$, $2.5e$-$4$; prior value, $p = 0.25$, $0.27$, $0.034$). **d** Comparison of GLM fitting across cortical regions. Δ Bayesian information criterion (BIC) shows the difference of model fitting between the full GLM and model without each task kernel (two-sided Wilcoxon signed rank test). Source data are provided as a Source Data file.

performance depended on sensory confidence. We further divided the trials by the difficulties of tone clouds (Supplementary Fig. 14c). The choice decoding was better in the easy than in the difficult stimuli. At the choice timing, the decoding performance was similar between the long- and short-sound trials in M2, while, in mPFC, the performance was slightly but significantly better in the long- than in the short-sound trials.

We then directly compared the decoding performance of 3 cortical regions by subsampling the number of neurons and trials in long-sound trials (Fig. 7d and Supplementary Fig. 13b). In each session, we randomly picked 100 neurons and 180 trials for 10-fold CV. This subsampled CV was repeated 1000 times. We found that the prior readout did not have significant differences among the AC, M2, and mPFC (two-sided Wilcoxon signed rank test, $p = 0.080$−$0.55$). In contrast, the sound and choice readouts were better in the AC and M2, respectively, than in the other two regions (sound: $p = 8.7e$-$9$ and $1.1e$-$6$; choice: $p = 2.7e$-$5$ and $2.8e$-$4$). These results were consistent when we subsampled the 50 neurons and 180 trials in each session (Supplementary Fig. 13b and 14d). These results suggest the global and localized readout of prior value and sound/choice, respectively.

We further analyzed how sound and choice decoding changed during sounds. In the AC, the sound readouts were improved by long sounds (Fig. 7e left and Supplementary Fig. 13c), suggesting a temporal integration of sound at the stage of AC. The sound readout from the AC was stable across blocks[5], while the readouts from the M2 and mPFC were biased by the blocks (Fig. 7e middle and right). The unbiased sound readout from the AC was important to compute the action value in our RL model (Fig. 2). The choice decoding of the M2 and mPFC was improved during long sounds (Supplementary Figs. 13c and 14e).

**Orthogonal representations of sounds and blocks in the AC and the global separation of blocks in the AC, M2, and mPFC**
Our decoding analyses proposed that (i) the AC had a stable sound readout and (ii) the AC, M2, and mPFC had a prior-value readout

before and during sounds. To elucidate these two points, we analyzed the state dynamics of blocks, sounds, and choices by introducing "Coding Direction" as was done in previous studies (Fig. 8a)[44,45]. This analysis utilized our task structure to compute the population activity vectors for separating the asymmetric-reward blocks (block mode) and for separating the sound category (sound/choice mode). As our tone-index analyses showed that the activity of mPFC and M2 neurons mainly represented choices while the activity of AC neurons represented sounds (Figs. 3, 5, and Supplementary Fig. 9), we used only the correct trials and made the separation of sound and choice categories identical. Thus, the sound/choice modes in the mPFC and M2 were mainly for choice, while it was for sound in the AC. Similar to the decoding analyses, we analyzed the sessions that simultaneously recorded the activity of more than 20 neurons. Before applying QR decomposition in the block and sound/choice modes for orthogonalization, we analyzed the cosines between the two modes. We found that the AC and mPFC had a better separation of the two modes than the M2 (Fig. 8b). The orthogonalized average state dynamics in the AC showed that the block and sound modes were separated throughout the trials (Fig. 8c). The explained variances of the two modes were better than chance in all the recorded regions (two-sided Wilcoxon signed rank test, $p = 5.4e$-$7$–$3.8e$-$6$) (Fig. 8d).

We further tested the orthogonality of block and sound/choice modes (Fig. 8e)[46]. In block mode, we compared the trajectories between choices; in the sound/choice mode, we compared the trajectories between blocks. In the AC, the neural trajectories did not have significant differences between sounds and blocks. On the other hand, the neural trajectories of one mode were affected by the other in the M2 and mPFC. These results suggest the orthogonal state space of blocks and sounds in the AC. Block modes were separated throughout the trials, especially in the AC and mPFC (Fig. 8e), suggesting stable block separations.

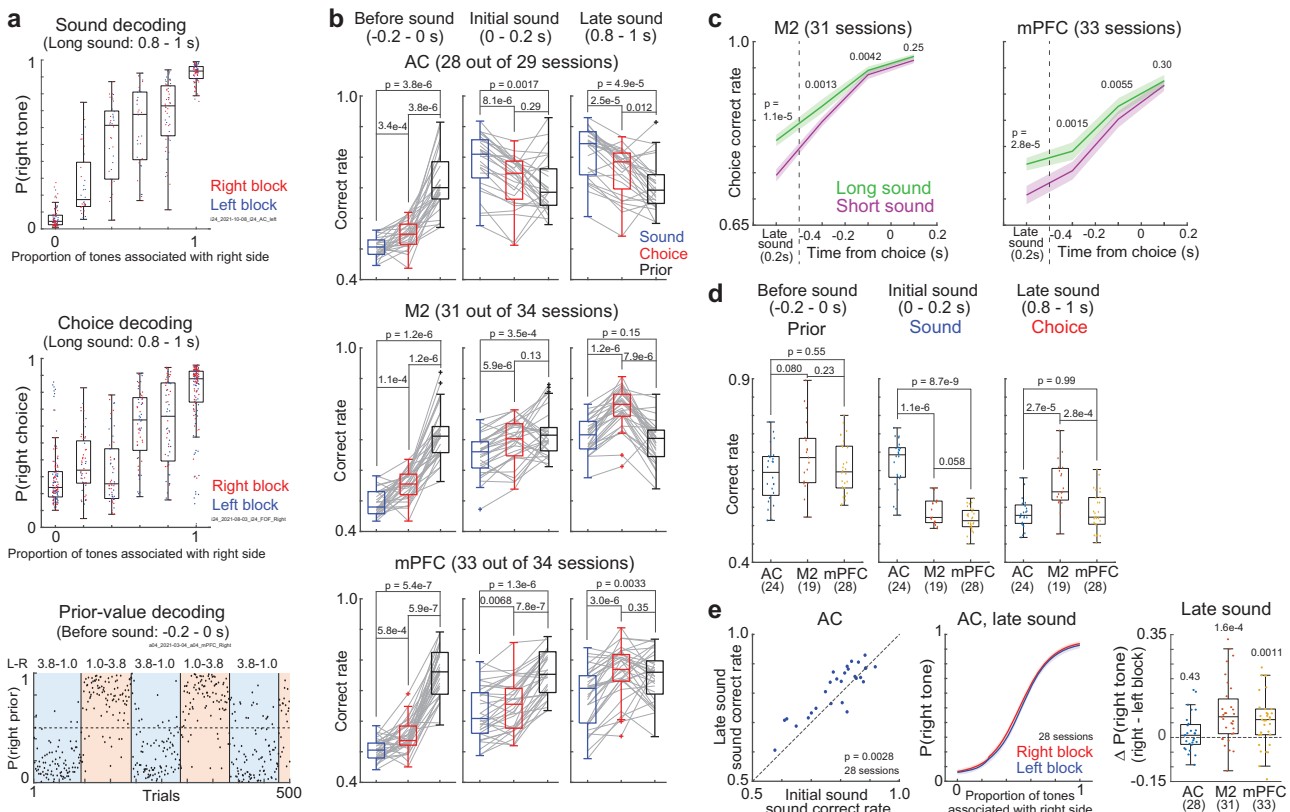

**Fig. 7 | Localized readout of sound and choice and global readout of the prior value from the auditory cortex (AC), secondary motor cortex (M2), and medial prefrontal cortex (mPFC). a** Sound, choice, and prior value decoding in example sessions. Top: sound decoding in an AC session. Dots show the probability estimate as a right-rewarded tone in each trial. Middle: choice decoding in an M2 session. Dots show the decoded right-choice probability (central and edges of the box: median, 25th, and 75th percentiles; whiskers: most extreme data points not considered outliers (beyond 1.5 times the interquartile range)). Bottom: prior-value decoding in an mPFC session. Dots show the decoded binarized prior value. **b** Comparison of decoding performance within each cortical region. Sessions with more than 20 neurons were analyzed (28, 31, and 33 sessions from the AC, M2, and mPFC) (*p*-value in the two-sided Wilcoxon signed rank test) (central and edges of the box: median, 25th, and 75th percentiles; whiskers: most extreme data points not considered outliers (beyond 1.5 times the interquartile range)). **c** Comparison of choice decoding performance between the long- and short-sound trials. Medians and robust standard errors (two-sided Wilcoxon signed rank test). **d** Comparison of

decoding performance across cortical regions. We subsampled the number of neurons and long-sound trials in each session. Sessions with more than 100 neurons were used (two-sided Mann–Whitney *U*-test) (24, 19, and 28 sessions from the AC, M2, and mPFC) (central and edges of the box: median, 25th, and 75th percentiles; whiskers: most extreme data points not considered outliers (beyond 1.5 times the interquartile range)). **e** Stable sound decoding in the AC. Left: Comparison of sound decoding performance between initial and end of long sounds (two-sided Wilcoxon signed rank test, 28 sessions). Middle: neurometric function of sound decoding. Means and standard errors (28 sessions). Right: stability of sound decoding. Δ p(right tone) was the difference of average neurometric functions between the right and left blocks. Parentheses show the number of sessions (*p*-value in the two-sided Wilcoxon signed rank test) (central and edges of the box: median, 25th, and 75th percentiles; whiskers: most extreme data points not considered outliers (beyond 1.5 times the interquartile range)). Source data are provided as a Source Data file.

## Discussion

Our results showed that mouse behavior in a tone-frequency discrimination task depended not only on the tone frequencies but also on the tone durations (Fig. 1): the mouse choices were less accurate and more biased toward the large-reward side in short-sound trials than in long-sound trials. In addition, the outcome experiences based on less accurate stimuli in terms of both the tone frequencies and durations affected the choices in the next trial compared to those based on accurate stimuli, consistent with a previous finding of sensory confidence-dependent value updating[13,34]. Our RL model suggested that the reward expectation of each choice (i.e., prior value) was modulated by sound inputs to compute action values (Fig. 2). Although we did not find brain regions representing the action values, we found that the prior values and choices were additively represented in the mPFC neurons (Fig. 4). The sounds and choices were selectively represented in the AC and M2, respectively (Fig. 6). In contrast, the prior values were decoded from all 3 recorded regions before and during sounds (Figs. 7 and 8). As the representations of sounds and choices were required within trials, while the prior values were needed

to be maintained across trials for future actions, our results suggest a localized and global computation of task variables for short- and long-time scales, respectively, in the cerebral cortex (Supplementary Fig. 15).

It has been reported that an RL framework based on a partially observed Markov decision process (POMDP) successfully models the ramping activity of neurons in the posterior parietal cortex (PPC) in a perceptual decision-making task[7,12], supporting that the RL model can apply to both perceptual and value-based decisions[9,13,47]. By utilizing the short-sound trials, our RL model estimated how the prior values were gradually modulated by sensory inputs to form the action values during long-sound presentations (Fig. 2). We used the model parameters of short-sound trials as the parameters at the initial phase of long sounds because the long- and short-sound trials had identical settings until the end of short sounds (Fig. 1). This analysis was supported by our neuron data in which the neural activity at the end of short sounds was correlated with the activity of the corresponding duration of long-sound stimuli, rather than the end of long sounds (Supplementary Fig. 5a). Mice temporally integrated the short sound

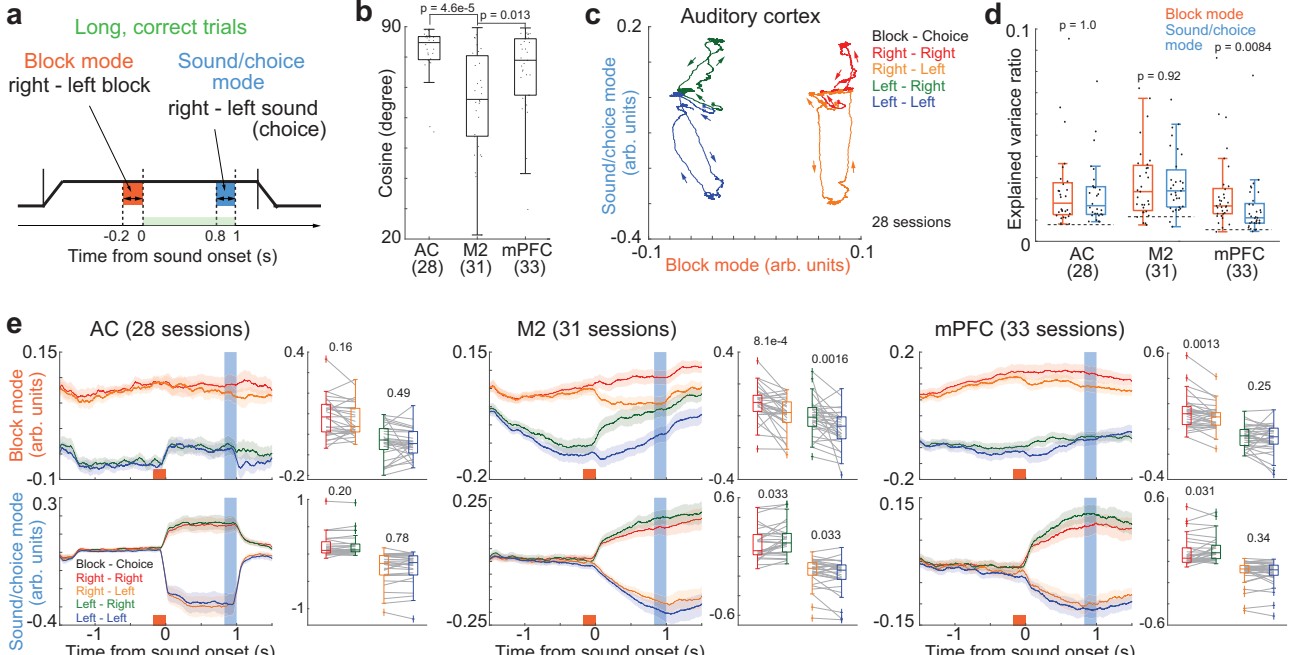

**Fig. 8 | State space analysis explains orthogonalized separations of block and sound in the auditory cortex (AC) and stable block separations in the AC, secondary motor cortex (M2), and medial prefrontal cortex (mPFC).** **a** Schematic of state-dynamics analysis. We defined the block mode and sound/choice mode based on the simultaneously recorded activity in each session. As we defined the modes from the activity in correct trials, the sound- and choice-separating modes were identical. **b** Separation of block and sound/choice modes. Parentheses show the number of sessions (two-sided Mann–Whitney *U*-test) (central and edges of the box: median, 25th, and 75th percentiles; whiskers: most extreme data points not considered outliers (beyond 1.5 times the interquartile range)). **c** Population trajectories in the AC averaged across sessions. **d** Explained variance ratio. Dotted lines show the baseline. Two-sided Wilcoxon signed rank test compared the ratio between the block and sound/choice modes (central and edges of the box: median, 25th, and 75th percentiles; whiskers: most extreme data points not considered outliers (beyond 1.5 times the interquartile range)). **e** Block and sound/choice modes in each block-choice combination. Left: means and standard errors. Right top: block modes were compared between choices at the sound end. Right bottom: sound/choice modes were compared between blocks (two-sided Wilcoxon signed rank test) (central and edges of the box: median, 25th, and 75th percentiles; whiskers: most extreme data points not considered outliers (beyond 1.5 times the interquartile range)). Source data are provided as a Source Data file.

pulses in the tone cloud and made accurate choices in long-sound trials compared to those in short-sound trials (Fig. 1), showing that the action values were computed even at the late phase of long sounds.

The action-value representations in the mPFC were reported in a previous study[13]. Our study found that the mPFC neurons additively represented the prior values and choices (Fig. 4 and Supplementary Fig. 8). Particularly, we found that the neurons activated before sound mainly showed the shifts of representations from prior values to choices, while the neurons activated only after sounds mainly represented choices (Supplementary Fig. 7). Compared to the mPFC, the M2 neurons had smaller prior modulations (Supplementary Fig. 10). Instead, the M2 had larger choice modulations and better choice decoding than the mPFC (Fig. 7d). The M2 had a similar choice decoding between the long and short sound trials at the choice timing (Supplementary Fig. 14c).

Consistent with previous findings[5,17], the activity of individual AC neurons was modulated by reward blocks by increasing or decreasing the activity at the preferred tone frequency (Supplementary Fig. 5c). By setting proper high-dimensional population vectors (Fig. 7), the AC generated block and sound readouts, consistent with a recent finding of the population readout from the visual and motor cortices[48]. We also found that the block- and sound-population vectors were orthogonal (Fig. 8), suggesting that other brain areas can receive sound frequency inputs from the AC irrespective of reward priors (Fig. 7e)[5]. The performance of choice decoding (Fig. 7d) and encoding (Fig. 6c) were similar between the AC and mPFC, possibly because of the choice-modulated sound representations in the AC (Supplementary Fig. 11a, b). The choices of mice were represented in M2 (Figs. 6, 7, and Supplementary Fig. 13). These results suggest that the sensory inputs

and choices were locally represented in the AC, and M2, respectively, with some modulations by prior values. These representations were required rapid changes in every single trials. A recent study utilizing Neuropixels 1.0 found that pre-action events, such as visual stimuli and upcoming choices, are represented in restricted regions in the neocortex[49], consistent with the localized computation in our study.

Our study found that the prior values were maintained in all the three recorded regions (Figs. 7 and 8). In the population level, we observed a clear prior modulation of activity in the mPFC and M2, but less clear in the AC (Figs. 3, 4, 5 and Supplementary Fig. 9), because the prior modulations of AC neurons were positive or negative in terms of the preferred tone frequency (Supplementary Fig. 5c). The prior-modulated neurons in the recorded areas were potentially enough to perform the prior-value decoding (Fig. 7), as previous studies show that even a single neuron can decode a task variable as well as the animal behavior[50–52].

The global brain-wide representation of prior is also reported by a previous study including the cortical and subcortical regions[53]. Another study finds a global representation of choices and task engagements[49]. In previous studies, the neural modulations by reward expectations are observed across sensory cortices[5,17], parietal[54], frontal[13,22], and motor cortices[10]. However, these studies were mainly focused on a specific brain area with different tasks, they were difficult to test whether the modulations were specific to the target area or globally distributed across a brain. Our study appreciated the Neuropixels probe and found that the prior-value representations were distributed across cortical areas. As the prior knowledge about the task is essential not only for executing action at the late stage of decision processing but also for modulating sensory processing[55] or for

predictive coding based on prediction errors[56–58], the global prior representations are possibly important for various steps of decision making.

Our main findings about global prior representations were based on the population of neurons across mice in task-relevant neurons (Fig. 6). Further experiments with neural manipulations are essential to investigate whether the prior encoding of early sensory processing is necessity to generate prior-value dependent choices. Also, precise experiments are required to distinguish the neural representations of priors with those of body movement[42], motor preparation, or attention[18], as these variables are known to modulate the activity[53].

To elucidate the localized and global computations in the cerebral cortex, it is important to test the role of the orbitofrontal cortex (OFC) and PPC. The OFC is known to represent sensory confidence[59], implying that the sensory inputs are once stored in the OFC for value computations. The PPC temporally integrates visual evidence for choices in a perceptual decision task[7]. In addition, the PPC is causally involved in the representation of prior knowledge of sensory or choice[40,54,60], implying that the PPC is involved in the integration of prior knowledge and sensory inputs[6]. It is also important to use female mice in our behavioral task to investigate the sex differences of neural circuits in decision-making.

Finally, our decoding analyses found that the AC improved the tone-frequency readout during long sounds (Fig. 7e), suggesting that sound evidence was temporally integrated at the stage of AC. This is surprising because in a visual perceptual decision-making task in monkeys, the temporal integration of visual stimuli occurs in the PPC, while the visual cortex mainly passes the visual information to the PPC[7,61]. This difference between the auditory and visual processes may come from different contributions of brain regions for decision-making. For example, the M2 but not PPC is necessary for choices in auditory perceptual decisions in rodents[25]. A feedback circuit from the PPC to the AC may enhance the frequency discrimination in the AC, as the AC-PPC circuit is involved in integrating sound evidence in gerbils[62].

In summary, we found that the cerebral cortex locally represented the sensory inputs and choices, which changed rapidly in single trials. In contrast, the mPFC, M2, and AC represented the prior values that were slowly updated and maintained across trials. These results suggest a localized and global computation of task variables used in short- and long-time scales, respectively.

## Methods

We used 8 male CBA/J mice (Charles River, Japan) that were 8–13 weeks of age at the start of behavioral training. All animal procedures were approved by the Animal Care and Use Committee at the Institute for Quantitative Biosciences (IQB), University of Tokyo. Mice were housed in a temperature-controlled room with an ambient temperature of 26.6 degrees Celsius, an average humidity of 23.3%, and a 12 h/12 h light/dark cycle. All experiments were performed during the dark cycle.

### Surgery

Surgical procedures and the original versions of the behavioral task were previously described[5,16,27]. Before surgery, the mice were restricted to 1.5 mL of water per day or had free access to citric acid water for at least one week. The weight of each mouse was checked daily to avoid dehydration. Surgery had two steps. The first step was to implant a custom-designed lightweight head bar for behavioral training. The second step was a craniotomy for electrophysiology. In both surgeries, mice were anesthetized with isoflurane (1.5% at induction, below 1% at maintenance) with an additional analgesic (meloxicam 2 mg/kg, subcutaneous) and eye ointment. The mice were placed in a stereotaxic apparatus. For head-bar implantation, the scalp was removed above the entire cortical area. The skull was cleaned with povidone-iodine and hydrogen peroxide. The head bar was attached to the skull with

superbond adhesive (Sun Medical or Parkell S380) and cyanoacrylate glue (Zap-A-Gap, PT03)[42].

Craniotomy was performed under isoflurane anesthesia. A tiny hole within 1 mm in diameter was made by drilling cyanoacrylate glue and the skull at a recording site to expose the brain surface. The recording site was chosen from the following candidates: the mPFC recording sites were centered at +1.8 mm anterior–posterior (AP), ±0.8 mm medio–lateral (ML) from bregma; the M2 sites were centered at +1.5 AP, ±1.0 ML; the AC sites were centered at −3.0 AP, ±3.7 ML (Supplementary Fig. 2). The M2 sites corresponded to the location of the frontal orienting field (FOF) in rats[26]. The dura mater was removed, and the hole was filled with a mixture of agar and PBS. The agar was coated with silicone oil (30,000 cSt, Aldrich) and sealed with Kwik-Sil (World Precision Instruments) to prevent the brain surface from drying.

### Tone-frequency discrimination task

After recovery from head-bar implantation, behavioral training started. The weight of each mouse was carefully monitored, and additional water was given after daily training to prevent dehydration. All the training was performed inside a sound-attenuating booth (O'hara Inc.) or a custom-made training box, while all the electrophysiological recordings were performed inside the sound-attenuating booth. Mice were head-fixed and positioned over a cylinder treadmill. One speaker (#60108, Avisoft Bioacoustics) was placed diagonally to the right of the mice for auditory stimulation. The speaker was calibrated with a free-field microphone (Type 4939, Brüel and Kjaer). Precalibrated water was delivered through two spouts connected to an infrared lick sensor (O'hara Inc.). The spouts were moved back and forth in each trial. The position of the spouts was controlled by an Arduino with a servo motor (hsb-9475sh, Hitech) and measured with a rotary encoder (E6A2-CS3C, Omron electronics). The behavioral system was controlled by a custom MATLAB (MathWorks) program running on the Bpod r0.5 framework (https://sanworks.io) on Windows.

Similar to our previous behavioral task[16], mice selected the left or right spout to receive a water reward depending on the frequency of the sound stimulus (Fig. 1). Each trial started by retracting the two spouts away from the mice. After a constant interval of 1.5 s, a sound stimulus of the tone cloud began. The tone cloud consisted of a series of 30 ms pure tones with rise/decay ramps of 3 ms at a rate of 100 tones per second[29,30]. The frequency of each tone was sampled from 18 logarithmically spaced slots (5 to 40 kHz). The tone cloud in each trial contained the low (5–10 kHz) and high-frequency tones (20–40 kHz) and was categorized as low or high depending on the dominant frequency. The proportion of high tones in each tone cloud was selected from 6 settings (0, 0.25, 0.45, 0.55, 0.75, or 1) with a probability of 25%, 12.5%, 12.5%, 12.5%, 12.5%, or 25% (i.e., 2:1:1:1:1:2). Although the proportion of high tones had low variability from the 6 settings except for the 100% low or high tone during the training phase, we used the exact 6 settings for the electrophysiological recording phase. In each setting, we used 20 varieties of tone clouds. The settings for short sounds were identical to the corresponding time window of long sounds.

After a 0.5 s delay from the sound end, the two spouts were moved forward, and the mice could select the left or right spout. The high- or low-category tone in a correct trial was associated with a reward of 2.4 µl of 10% sucrose water in either the left or right spout. The association between the tone category and rewarded choice was determined for each mouse. The selection of the opposite spout triggered a noise burst (0.2 s), marking an error trial. The outcome was delivered immediately after the choice. When mice did not select a spout within 15 s from the start of the trial, a new trial started.

A duration of either a long (1.0 s) or short sound was randomly selected for each trial, except that the first 40 trials in each session were always long stimuli with 100% low- or high-tone clouds. The duration of the short sound was 0.2 s for 5 mice and 0.4 s for 2 mice. In the other one mouse, the short sound had a duration of 0.4 s at the

initial session of electrophysiology, and it was 0.2 s for the remaining 4 sessions. During the training phase of behavioral task, we gradually shorten the duration of short sounds. In some mice, the sound duration of 0.2 s was difficult to make the tone-frequency-dependent choices. For these mice, we set the duration of short sounds as 0.4 s. The intensity of the tone cloud was constant in each trial but was sampled from either 60, 65, or 70 dB SPL (sound pressure level in decibels with respect to 20 µPa). After the first 40 trials, the reward amounts of the left and right spouts were switched by a block of 90–130 trials. In each block, the reward amount for a correct choice of a large–small reward pair was selected from 2.8–2.0 µl to 3.8–1.0 µl in each session. We defined the left and right blocks depending on the large-reward side. If mice did not complete 4 asymmetric-reward blocks, we did not use the session for analysis. After the behavioral task on each day (session), we passively presented the mice with sound stimuli to record the sound-responsive activity of neurons. Data for passive sound presentations are not shown in this manuscript.

### Electrophysiological recording with Neuropixels 1.0 and spike sorting

Electrophysiology was performed with Neuropixels 1.0 (IMEC) based on the procedure in a previous study[63]. Every day, i.e., every session, of the behavioral task, we inserted one or two Neuropixels probes. For the mPFC recordings, the probe was tilted 8 degrees in the medial direction except in one mouse (0 degrees). For the AC recordings, the tilted angle was set to 20 degrees in the lateral direction. The angle was 0 degrees for the M2 recording. To identify the probe location in the post hoc fixed brain, the probe was soaked with a diluted solution of CM-DiI or DiA (ThermoFisher Product #V22888 or #D3883). The probe was manually lowered to the brain surface through a mixture of agar and PBS. The brain surface was defined at the depth when spikes were first observed at the recording electrodes around the tip during insertion. We then inserted the probe at a speed of 120 µm/min to prevent damage to the brain tissue[64]. Recording was started at least 5 min after the end of probe insertion.

The head bar and tip electrode of the Neuropixels probe were used as the ground and reference, respectively. The 384 electrodes from the tip were used for recording. Open-Ephys GUI acquired the neural data at a sampling rate of 30 kHz with a gain of 500 (PXIe acquisition module, IMEC)[65]. The task events, including treadmill rotations, were sampled at 2.5 kHz (BNC-2110, National Instruments). After recording, the probe was slowly extracted, and the probe-inserted hole was covered with Kwik-Sil. We inserted the probe 1 to 4 times (i.e., 1–4 days) in each hole in both the left and right hemispheres of the brain.

For spike sorting, the recording data during the behavioral task and passive sound presentations in each day from one mouse were concatenated. Spike sorting and manual curation were performed with Kilosort 3 in MATLAB (https://github.com/cortex-lab/KiloSort) and Phy in Python (https://github.com/cortex-lab/phy).

Kilosort-3 spike sorting tracked the approximate depth of spikes from each unit during a session. We defined the depth of each unit from the approximate location of the probe and the electrode position, which measured the maximum amplitude of spikes on average. For the Neuropixels probe for M2 recording, we used the units for analysis when the estimated spike depth from the brain surface was above 1.5 mm. For the mPFC recording probe, we used the units for analyses when the spike depth was (i) above 3.2 mm for the 8-degree tilted probe (7 mice) and (ii) above 3.0 mm for the 0-degree probe (1 mouse). We analyzed all the recorded units for the AC probe. For the PPC recording probe, we defined the units from PPC when the spike depth was above 1.0 mm.

### Histology

After electrophysiological recording, the mice were perfused with 10% formalin solution. Mice were deeply anesthetized with an induction of 5% isoflurane and further with a mixture of 0.3 mg/kg medetomidine, 4.0 mg/kg midazolam, and 5.0 mg/kg butorphanol. The brains were preserved in 10% formalin solution for 24 h and stored at 4 °C in PBS. The brains were sectioned by a vibratome (VT1000S, Leica Biosystems) along the coronal plane with a thickness of 100 µm. The slice tissues were mounted on glass slides with DAPI mounting medium (Vectashield). Images were acquired with a confocal laser scanning microscope (FV3000, Olympus) or a fluorescence microscope (BZ-X710, Keyence) under x4 magnification (Supplementary Fig. 2).

### Data analysis

All analyses were performed in MATLAB (MathWorks), except the manual curation of spike-sorting data that was performed in Python. On each day of the session, trials in which mice succeeded in selecting the left or right spout were analyzed. We analyzed the behavioral data from 8 mice in 122 sessions (mouse i01 – i08: 23, 12, 11, 21, 20, 20, 10, and 5 sessions). We did not use the sessions for analyses (i) when mice did not select the correct sides at least 75% of the time for the 100% low- and high-tone clouds in long-sound trials (7 sessions) and (ii) when mice did not complete the 4 asymmetric-reward blocks (2 sessions). Among the 122 sessions, the neural activity was recorded from the AC from 7 mice in 29 sessions (mouse i01 – i08: 5, 1, 0, 4, 7, 7, 4, and 1 sessions), the M2 from 8 mice in 34 sessions (6, 6, 6, 6, 3, 4, 1, and 2 sessions), and the mPFC from 8 mice in 34 sessions (8, 5, 5, 3, 3, 4, 4, and 2 sessions). The other 32 sessions (7 sessions were simultaneously recorded with AC) targeted the posterior parietal cortex (PPC). However, as the number of neurons from PPC was less than half on average per session compared to the other 3 regions, we did not use the PPC data for analyses.

### Behavioral analysis

We used logistic regression (MATLAB: glmfit) to quantify which tone-cloud timings and asymmetric-reward blocks were correlated with the choice in each trial (Fig. 1e). The detailed method for the psychophysical kernel was reported in previous studies[16,31–33]. In the short sound trials, we divided the tone clouds into the former and latter half and investigated how the tone frequencies of former and latter sounds and the asymmetric-reward blocks were correlated to the choices. In the long-sound trials, we divided the tone clouds with 5-time windows with each 0.2 s. This analysis only used trials with moderate (75% low or high frequency) and difficult (55% low or high) tone clouds.

We also tested how the reward history of each choice in previous trials was correlated to the choice in the current trial (Supplementary Fig. 1c)[60]:

$$P(\text{right},t) = \int_{d+\text{reward\_bias}(t)}^{1} ZN\left(x | E_{\text{right}}(t), \sigma^2\right) dx$$

$$\text{reward\_bias}(t) = \frac{\exp\left[\sum_{i=1}^{T}\{\beta_i \text{LeftReward}(t-i)\}\right]}{\exp\left[\sum_{i=1}^{T}\{\beta_i \text{LeftReward}(t-i)\}\right] + \exp\left[\sum_{i=1}^{T}\{\beta_i \text{RightReward}(t-i)\}\right]}$$

$$(1)$$

$p(\text{right},t)$ was the rightward choice probability at trial $t$. $E_{\text{right}}(t)$ was the proportion of tone frequency associated with a rightward choice in a tone cloud. $Z$ truncated the Gaussian distribution between 0 and 1 here and throughout. $d$ and $\sigma$ were the decision threshold and perceptual uncertainty. For simplicity, we used the same $d$ and $\sigma$ for the long and short sound trials in this analysis. reward_bias$(t)$ biased the decision threshold based on the reward history. $\beta_i$ was a regression coefficient. LeftReward and RightReward were the received reward amount in the left and right choice, respectively. $T$ was the number of past trials. We investigated whether the additional past trials $T$ from 1 to 10 increased the prediction accuracy of model with the likelihood ratio test[66] ("Methods" section, "Model comparison" section).

We modeled the choice behavior of mice with a psychometric function for which the equations were reported in our previous

study[16]. In summary, the psychometric function modeled the stimulus sensitivity of mice with a truncated Gaussian ranging between 0 and 1[59]. Our model investigated whether the choice biases and stimulus sensitivities of mice depended on the asymmetric-reward blocks and stimulus durations. We also tested whether a lapse rate was necessary to model the choice behavior by the likelihood ratio test[66] (Supplementary Fig. 1d).

### Reinforcement learning model

We analyzed the choices the mice made with a reinforcement learning (RL) model[9,12,13,16]. The RL model had the stored expected reward of left and right choice $V_a$ in each trial $t$, defined as the prior value. The model used the prior value to compute the decision threshold $x_O$ with a softmax equation and an inverse temperature parameter $\beta$:

$$x_0(t) = \frac{\exp(\beta V_{\text{left}}(t))}{\exp(\beta V_{\text{left}}(t)) + \exp\left(\beta V_{\text{right}}(t)\right)} \quad (2)$$

The softmax equation modeled a perceived reward size that might be different from the actual amount of water[67,68]. The choice probability was estimated from duration-dependent perceptual uncertainty $\sigma_s$ and a bias parameter $d$:

$$P(\text{right},t) = \int_{x_0(t)+d}^{1} ZN(x|E_{\text{right}}(t),\sigma_{s(t)}^2)dx \quad (3)$$

$E_{\text{right}}(t)$ was the proportion of tone frequency associated with a rightward choice in a tone cloud. Independent from the choice probability, the perceived probability of left- and right-category tone, or the probability of left- and right-rewarded context, was estimated as follows:

$$P(\text{left category},t) = \int_0^{0.5} ZN\left(x|E_{\text{right}}(t),\sigma_s^2\right)dx$$
$$P(\text{right category},t) = \int_{0.5}^1 ZN\left(x|E_{\text{right}}(t),\sigma_s^2\right)dx \quad (4)$$

Because of $\sigma_s$, we were able to estimate the perceived tone category at the duration of short sounds even in the long-sound trials. The perceived category corresponded to the belief about context which was either the left- or right-rewarded. The action value $Q_a$ was defined by multiplying the prior value $V_a$ with the associated context, i.e., tone category $P(\text{category},t)$[13], as the left- or right-category tone was only associated with the reward on one side. We defined $Q_a$ as follows:

$$Q_{\text{left}} = P(\text{left category},t)V_{\text{left}}$$
$$Q_{\text{right}} = P(\text{right category},t)V_{\text{right}} \quad (5)$$

In addition to the multiplicative action value, we used the additive action value $(P(\text{category},t) + V_a)$ for modeling mouse behavior. The average Bayesian Information criterion (BIC) per mouse of multiplicative model (521.883) was smaller than that of additive model (523.233), showing that the multiplicative RL model fit to the behavior. We updated the stored prior value $V_a$ with forgetting Q-learning, which used the error $\delta$ between the actual outcome $r(t)$ and action value[9]:

$$V_a(t+1) = \begin{cases} V_a(t) + \alpha\delta & \text{if } a = a(t) \\ (1-\alpha)V_a(t) & \text{if } a \neq a(t) \end{cases} \quad (6)$$

where $\alpha$ was the learning rate. $\delta$ was defined as $r(t) - 2Q_a(t)$. We used forgetting Q-learning, which fit better in the choice behavior of rodents than a standard Q-learning model in previous studies[47,68,69]. We tested whether the model using the action values fit the choice behavior compared to that using only the prior values in our previous study (Supplementary Fig. 1g)[16]. The initial prior value for each choice was the average amount of expected reward (i.e., 1.2).

### Model comparison

For model fitting of psychometric functions and RL models, we defined the likelihood $l(t)$ from the estimated choice probability in each trial (e.g., Eq. 3):

$$l(t) = \begin{cases} P(\text{right},t) & \text{if } a(t) = \text{right} \\ 1 - P(\text{right},t) & \text{if } a(t) = \text{left} \end{cases} \quad (7)$$

We then analyzed the likelihood in each session $L$ using the trials during the left and right blocks:

$$L = \prod_{t=1}^{T} l(t), \quad (8)$$

where $T$ was the number of trials. Model comparison within the psychometric functions or the RL models was performed using the likelihood ratio test[66], which investigated whether an additional parameter in the model significantly increased the averaged log-likelihood per session ($p < 0.05$) (Supplementary Fig. 1c, d, e). The model parameters were fit to achieve the maximum likelihood. We used the Bayesian information criterion (BIC) to compare the performance between the psychometric function and the RL model (Fig. 2c)[70]:

$$\text{BIC} = -2\log(L) + k\log(T) \quad (9)$$

where $k$ was the number of free parameters.

### Spike data analysis

We first identified task-relevant neurons that significantly increased the activity during the task ($p < 1e-10$ in the one-sided Wilcoxon signed rank test). The activity of each neuron was aligned at the sound onset or at the choice timing. For sound-aligned activity, the task-relevant neuron increased the activity compared to the baseline activity in at least one-time window (0.1 s) between −1.5 and 2.5 s for the long-sound trials (40 windows) or between −1.5 and 1.7 s for the short-sound trials (32 windows). For the choice-aligned activity, the task-relevant neuron increased the activity in at least one-time window between −0.5 and 2.5 s from the choice onset either for the long- or short-sound trials ($30 \times 2 = 60$ windows, in total $40 + 32 + 60 = 132$ windows). The baseline for sound-aligned activity was the activity at −0.2–0 s from spout removal, i.e., before trial initiation. The baseline for choice-aligned activity was the activity at −0.2–0 s from the spout approaching, i.e., between the end of the sound and making a choice. In the PPC, number of task-relevant neurons was only 22 on average per session which was less than one-third of those from the AC, M2, and mPFC (70–94 neurons on average, two-sided Mann–Whitney U-test, $p = 3.5e-7–1.7e-6$).

In the task-relevant neurons, we analyzed the tone index, which was the difference between the average activity of left- and right-category-tone trials in a time window of 0.1 s:

$$\text{Tone index} = \frac{\text{mean(spikes(right tone trials))} - \text{mean(spikes(left tone trials))}}{\text{mean(spikes(right tone trials))} + \text{mean(spikes(left tone trials))}}$$
(10)

The tone index ranged between −1 and 1 and was independently analyzed for correct and error trials (Fig. 3d). We also analyzed the prior index, which was the difference in average activity between the left- and right-prior-value dominant trials ranging between −1 and 1 (Supplementary Fig. 5b). To investigate the performance of choice and sound decoding in single neurons, we analyzed the area under the receiver operating curve (auROC)[39]. For choice decoding, we compared the activity distribution between the left and right choice trials. For sound decoding, we compared the distribution between the low- and high-category-tone trials. The range of auROC was between 0.5 and 1 for the random and perfect classification.

We defined the preferred sides (left or right choice) of task-relevant neurons. The preferred side was defined once based on the activity during the long-sound correct trials at the time window with the largest activity difference from the baseline (one-sided Wilcoxon signed rank test). The average activity on the preferred side was higher than that on the nonpreferred side. For example, when the average activity in correct left-choice trials was higher than that in correct right-choice trials, the preferred side was defined as left. The preferred tone and preferred block were defined based on the preferred side. The preferred tone was associated with the reward on the preferred side. The preferred block had a large-reward amount on the preferred side.

### Encoding analysis with a generalized linear model (GLM)

Before and during the sounds, we investigated which task parameters of choices, sound frequencies, prior values, or action values were represented in single neurons (Supplementary Fig. 8b). The neural activity of 0.1 s was averaged, and log scaled, $y_{window}$. In the mPFC and M2, we analyzed whether the activity of each neuron was modulated by choices, prior values, or action values in each time window of 0.1 s with a following linear regression[8–10]:

$$y_{window}(t) = \beta_0 + \beta_1 RunningSpeed(t) + \beta_1 choice(t) + \beta_2 V_{relative}(t) + \beta_3 Q_{relative}(t) \tag{11}$$

where $t$ was the trial. RunningSpeed was the rotation of the treadmill in the corresponding time window. choice was the mouse's choice. $V_{relative}$ and $Q_{relative}$ was the relative prior value and action value of right side compared to the left ranging between 0 and 1. In the AC neurons, we analyzed whether the neurons represented sounds, prior values, or action values:

$$y_{window}(t) = \beta_0 + \beta_1 RunningSpeed(t) + \beta_1 E_{right}(t) + \beta_2 V_{relative}(t) + \beta_3 Q_{relative}(t) \tag{12}$$

where $E_{right}$ was the proportion of tone frequency associated with a rightward choice. In all the three cortical regions, we first analyzed the root mean squared error of model only using $\beta_0$ and $\beta_1$ (RunningSpeed) with 10-fold cross-validation (CV). The CV was repeated 10 times to reduce the noise from randomization. We then investigated whether the additional task parameters in the model reduced the squared error in CV (Δ root mean squared error; Supplementary Fig. 8b).

We also analyzed the neural encoding using entire time series of activity (Fig. 6). The neural activity was averaged every 0.1 s and smoothed with a Gaussian filter ($\sigma = 0.25$ s). We modeled the smoothed log-scaled activity with a generalized linear model (GLM)[9,45,49]. We used the MATLAB software package glmnet (https://glmnet.stanford.edu/index.html) for the analysis[41]. Our GLM had an equal number of kernels for the sound, choice, and prior value in the RL model, as the sounds and prior values were categorized based on the dominant tones and sides, respectively. In each of the 3 task variables, we prepared 28 time-locked kernels: 18 time-locked kernels in the long-sound trials between −0.2 and 1.5 s from the sound onset, and 10 kernels in the short-sound trials between −0.2 and 0.7 s from the sound onset. The total number of task-variable kernels was 168 (3 task variablesx2 categoriesx28 time windows). We also prepared the time-locked kernels for the choice-outcome combinations between −0.1 and 1.0 s from the choice onset (4 combinationsx12 time windows = 48 kernels) and for the running speed of the mouse analyzed from the treadmill rotation (1 kernel). In total, we prepared 218 kernels, including the constant term (168 + 48 + 1 + 1 = 218).

Our GLM fit the log-scaled neural activity with the elastic net with 99% L2 and 1% L1. The regularization parameter $\lambda$ was determined to minimize the error of 10-fold CV (cvglmnet). The CV split the data by trials and did not randomly split by time points to avoid contamination between the smoothed neural activity of training and test data. With

the estimated $\lambda$, we analyzed the regression coefficients of GLM using all the data. By comparing the log-likelihood between the full GLM and the model without sound, choice, or prior-value kernels, we analyzed the proportion of neurons encoding each task variable ($p < 0.01$ in the likelihood ratio test) (Fig. 6c). We also analyzed the difference in BIC between the full model and the model without each task-variable kernel (Δ BIC) (Fig. 6d).

### Decoding analysis

We decoded the sound category, choice, and prior category (i.e., the dominant side of the prior value) from the simultaneously recorded neural activity in each session. This decoding analysis used sessions with more than 20 neurons. We used a sparse logistic regression in glmnet (https://glmnet.stanford.edu/index.html) for the analysis. We first compared the decoding performance of 3 task variables within a session by using all the recorded neurons and 10-fold CV (cvglmnet) with a time window of 0.2 s. In the decoding analysis, 10-fold CV was performed within the training data to separate all the parameter estimations, including the regularization parameter, from the test data (nested CV). The trained decoder then predicted the task variables in the test data. These CVs were performed 100 times to reduce the noise from randomization. We then analyzed the neurometric function by fitting the binarized decoding outputs with the psychometric function (Fig. 7e) ("Methods" section, "Behavioral analysis" section)[16].

To compare the decoding performance across cortical regions, we aligned the number of neurons and trials by subsampling 100 neurons and 180 trials in each session (Fig. 7d). This analysis used sessions with more than 100 neurons. We also subsampled 50 neurons and 180 trials in each session to validate the analysis (Supplementary Fig. 14d). We repeated the subsampled 10-fold CV 1000 times. To remove the crosstalk across decoders with 3 task variables, the performance of sound decoding was independently analyzed for the left- and right-choice trials and averaged. Prior decoding was also independently performed for the left- and right-choice trials. Choice decoding was performed every 4 tone-cloud-setting trials with moderate or difficult tone clouds, and the performances were averaged.

### State space analysis

We used "Coding Direction" analysis to investigate the population vector for separating neural activity between blocks (Block mode) and the population vector for separating activity between sound categories (Sound/choice mode) (Fig. 8)[44,45]. As this analysis only used the long-sound correct trials, the separations of sound- and choice categories were identical. We utilized our task structure to separate the activity between the block and sound/choice modes. The block mode used the activity between −0.2 and 0 s from the sound onset and analyzed the average difference in z-scored activity between the left and right blocks: $W_{block}$. The sound/choice mode analyzed the average activity difference between the left- and right-choice trials at the end of a long-sound (0.8–1.0 s from sound onset): $W_{sound/choice}$. To orthogonalize the block and sound/choice modes, we used QR decomposition[44,45]:

$$W = QR \tag{13}$$

where $W$ consists of the columns of $W_{block}$ and $W_{sound/choice}$. The first two columns of $Q$ were the orthogonal subspace of $W_{block}^{\perp}$ and $W_{sound/choice}^{\perp}$. Similar to the decoding analyses, we used sessions with more than 20 neurons.

**Quantification and statistical analysis.** All analyses were performed with MATLAB (MathWorks), except the manual curation of spike-sorting data with Python. Model fitting of psychometric functions and RL models was performed using the likelihood ratio test[16,66] (Supplementary Fig. 1) and Bayesian information criterion (BIC) (Fig. 2)

("Methods" section, "Model comparison" section)[70]. Behavioral data analyses were performed using the linear mixed-effects model (MATLAB: fitlme)[16], as we analyzed multiple sessions from each subject (mouse). To compare samples in each session, the fixed and random effects were the samples and subjects, respectively. We used the chi-square test to compare the proportion of neurons encoding task variables (Fig. 6c). We mainly used two-sided nonparametric statistical tests. Descriptions were made for one-sided statistical tests. Error bars for means are standard deviations or standard errors. Error bars for medians are median absolute deviations (MADs) or robust standard errors (1.4826*MAD/sqrt($n$); $n$=number of data points)[71,72]. All the statistical details are in the figure legends, figures, or results.

### Reporting summary

Further information on research design is available in the Nature Portfolio Reporting Summary linked to this article.

## Data availability

Source data are provided with this paper. The spike data generated in this study and Source data are deposited in Figshare [https://doi.org/10.6084/m9.figshare.24319705]. The raw electrophysiological data, which are not deposited because of the large data size (>1 TB), are available from the corresponding author upon request. Source data are provided with this paper.

## Code availability

Codes which support findings in our paper are available on GitHub [https://github.com/funamizu-lab/Ishizu_et_al_2023]. Codes and the intermediate source data are also available on Zenodo [https://doi.org/10.5281/zenodo.10881334].

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

## Acknowledgements

This work was funded by JSPS LEADER, Kakenhi (JP20K23339, JP21H00187, JP21H05243, and JP22H04766), the Senri Life Science Foundation, and the Japanese Neural Network Society (JNNS) for A.F.; JSPS Kakenhi (JP20K23317) for K.I.

## Author contributions

K.I. collected and analyzed the data. S.N. and Y.U. analyzed the data. A.F. designed the experiment, analyzed the data, and wrote the paper.

## Competing interests

The authors declare no competing interests.
