## [Peer Review File · Nature Communications]

Localized and global representation of prior value, sensory evidence, and choice in male mouse cerebral cortexREVIEWER COMMENTS

Reviewer #1 (Remarks to the Author):

The paper titled "Localized and global computation for integrating prior value and sensory evidence in the mouse cerebral cortex" investigates how the brain integrates prior knowledge of action outcomes and sensory evidence to compute action values. The integration of prior value and sensory inputs in the brain is a broad and fundamental topic for neuroscientists, and the combination of their original behavioral setup, large-scale neuronal recordings in three cortical regions, and sophisticated computational analyses can potentially provide significant insights for understanding this issue. Overall, the study is very interesting and informative, and the results are presented in a clear and structured manner, making them easily understandable. However, I believe that further refinement is needed to improve the clarity of the research objectives, the interpretation of results, and the discussion of limitations. Addressing these points would strengthen the paper and contribute to the existing knowledge in the field of decision-making neuroscience.

Major points:

1) The interpretation of global representations of prior value

The major finding of the global representations of prior value is only discussed in a single sentence as below, and the significance of this finding is not clear at all. Elaborating and discussing this, as well as other possibilities and limitations, are crucial.

"As the prior values need to be maintained across trials for future actions, such long maintenance may need global coding (lines 416-417)."

1-1) I could not find explicit information regarding the "maintenance of prior." How many trials are required to switch across different blocks? How many past trials are counted for current choice information? Basic information like this is necessary to evaluate the maintenance of prior information for individual animals.

1-2) Reward/punishment is normally represented brain-wide, and what they call the "prior value" may simply be equivalent to sustained reward experience. Are there mPFC neurons that shows sustained activity in response to correct/error outcome to subsequent trials? If there are, what is the relationship between those neurons and the neurons analyzed in Fig.3G?

1-3) An alternative possibility for this prior representation is movement-related activity. Do the authors record videos of mice behaviors on the treadmill during the task? It is known that task-irrelevant movement significantly influences activities in cortical areas (e.g., Musall et al., Nat. Neuro 2019). The previous reward experience may change their running speed, body orientation, and postural biases, allowing them to optimize their bodies for the following choices. Such biased motor activities might transmit globally to the brain, creating a correlation with prior value and biased choices in subsequent trials.

2) "Integration" is not directly evaluated

Despite their paper's title, "integration of prior and sensory inputs" are not clearly evaluated at the neural level under their model variables. According to their model, they have trial-by-trial value of $\text{conf_}Q_a$, which is defined by multiplying the prior value Q_a with associated tone category Prob. I think that evaluating neuronal activities across areas with these model-based latent variables are one of straightforward way to evaluate their claim and directly comparable with a previous study (Lak et al., 2018).

3) Sequential activity

"The confidence-value representations in the mPFC were reported in a previous study¹². We advanced the finding by that a temporal sequence of population neuronal activity in the mPFC modulated the prior-value representations with sensory inputs to compute the confidence values (Figure 3). (line 397)"

"we found that a temporal sequence of neurons in the medial prefrontal cortex of mice updated the prior reward expectations with incoming sensory inputs (lines 23-25)"

"This gradual modulation was represented in a sequential population activity of mPFC neurons (Figure 3). (line 374)"

3-1) Though they signify the finding of the "temporal sequence" in mPFC, its meaning is not discussed well. How is this sequential recruitment of neurons relevant to the integration of prior and sensory inputs?

3-2) They state that they first found that the task-relevant neurons were sequentially recruited during the long-sound trials, rather than mainly observing the ramping activity of neurons toward the end of sounds. However, as they picked up neurons based on the significant increase of neurons at one of the 132 time-windows from baseline, there is no wonder to observe "sequential recruitment" of cells (even random data can create a pseudo sequence). Did the authors cross-validate the sequential recruitment by randomly splitting the trials, particularly the ones activated during the tone presentations?

3-3) Since they claim that they found task-relevant neurons typically show peak activity at some point during the tone presentation and not a ramping type, it is more reasonable to show such transient neurons as examples rather than ramping-type neurons in Fig.3C.

3-4) The authors showed temporal sequence activity in Fig.3B and gradual integration of prior and sensory inputs in Fig.3GH, but there is no direct connection between the temporal sequence and the gradual integration, despite their claim that the temporal sequence updated the prior reward expectations as mentioned above. Do those neurons newly recruited during the sound presentation actually show similar tuning curves as in Fig.3EH? Alternatively, do those choice-preference neurons they analyzed in Fig 3G-H actually construct sequential population activity (like Fig.3B)?

4) Decision confidence

The use of the term "confidence" is confusing and not justified.

4-1) First, their use of variable names is not coherent (e.g., action value, Q value, conf_Q value, sensory confidence, confidence). Please make sure they are coherent.

4-2) Second, the use of the term "confidence" is different from that in the previous study (Lak et al., 2018), despite authors utilizing a very similar model. The authors express confidence as the multiplicative reward expectation, whereas Lak et al. describe confidence as a probability estimate of reward (sensory confidence). Utilizing the term "confidence" for their conf_Q value is the authors' original invention and not justified.

The original paper by Kepecs et al. (Nature 2008) uses confidence as the probability of the "correctness" of the choice given stimulus. Lak et al. follow this convention, showing that their fitted values of "confidence" follow the original signature of decision confidence (such as Fig.5b in Kepecs et al., Nature 2008, Fig.1K in Lak et al., Neuron 2018). Because it is not obvious that this kind of licking choice paradigm requires the computation of "decision confidence" for a post-decision wagering strategy, it is necessary to confirm such a confidence signature of the previous studies. Alternatively, I would suggest that they stick to objective and general terms such as stimulus difficulty and action value.

5) Individual variability of animals

The manuscript does not provide much information about the replicability of key findings in terms of different animals.

For instance, Figure 1C shows an example of a session that shows a sigmoid-type PMF, but it is not clear how much individual differences exist in their dataset. It would be helpful to show the psychometric curves with overlaid data from each subject or fitted values from individual animals.

The core finding of Figure 5C also replicable in at least multiple animals?

Minor:

1) I think that the first paragraph of the results section would be more appropriate to be merged into the last paragraph of the introduction.

2)

“These studies suggest a hypothesis that the cerebral cortex, including the sensory and frontal cortices, are orchestrated to integrate prior knowledge and sensory stimuli, although the role of each cortical region is still unclear.” (lines 59-61)

It is stated that "the role of each cortical region is unclear," despite there being multiple previous studies that show the combined representation of prior and sensory inputs in different brain regions.

“A recent study reported that the mPFC represents a confidence value that combines prior values and sensory inputs, although it is unclear where and how prior values and sensory inputs are represented and integrated in the brain. “(lines 54-55)

This sentence is also repetitive. While the authors state that Lak et al. showed "confidence value combines prior value and sensory inputs in mPFC," the latter half of the sentence still states "it is unclear where and how." Authors should elaborate on what is missing in the previous studies.

3) Figure 3: The authors described that they removed neurons recorded from PPC in the results section. However, since the reasoning is only given qualitatively, the necessity to remove the data from the analysis is not convincing. Given that this paper claims that prior information is represented "globally" across cortical areas, it is curious to know whether this finding is generalizable to PPC. Adding PPC analysis only for the Fig.3 encoding analysis will further support their claim (no need to include it in the paper if they are not comfortable). Alternatively, please justify why they could not include the PPC data in a quantitative way.

4) Figure 5C:

... Thus, the sounds and choices were encoded in the AC and M2, respectively, while the prior values were encoded in all 3 recorded regions. (lines 293-294)

This statement is not accurate for describing their results. What they found is that the relative proportion of neurons encodings for sounds and choices differs across areas, and there are no significant differences in the encodings of prior values among the three areas (I mean it is not like “X is encoded in A and Y is encoded in B”).

Reviewer #2 (Remarks to the Author):

This paper investigates how cortical activity is related to stimuli, choices, and reward value in the auditory (AC), secondary motor (M2), and medial prefrontal (mPFC) cortex. The authors were able to examine the interaction between these features by combining an auditory sensory discrimination task with perceptual uncertainty and block-based reward biases. Behaviorally, they demonstrate that animals combine stimulus information with reward block history to produce value-based choices. From their neural recordings, they show that sound-related activity is predominantly in the AC, choice-related activity is predominantly in M2 and mPFC, and all three areas contain activity related to reward bias block. While the manuscript would benefit from certain clarifications, it provides interesting insight that block-related activity exists in all three areas, and that this activity is affected by choice in M2 but not in AC or mPFC.

Specific comments:

In Figure 1E, “former” and “latter” are not defined, and the methods section references another paper for further explanation. Please define this in the figure legend, and include at least a summary of methods which are more fully explained in previous papers.

In Figure 3A, this is presumably only long sound trial data, though this isn’t stated. Please state this in the figure legend.

In Figure 3B and related text, it’s unclear whether there is a true sequence of activity spanning the sound in the mPFC, as opposed to a group of cells which are generally preferentially active during that period. One graphical aspect of this is the yellow stripe through Figure 3B (and other heatmaps) – this is present by definition since activity is z-scored and sorted, the authors might try sorting by one half of the trials and plotting the other half of trials to provide a clearer idea of how robust the sequence is. The Δ

activated neurons plot also shows a bump at time 0, which could indicate a small fraction of consistently active cells from tone onset rather than a ramp. The authors also state in line 252 that AC activity did not ramp “in contrast with the mPFC” (line 252 – possibly referring to the ramping of the orange line in Fig 3E?), making it less clear whether there is a ramp or not. This is all a minor point since the sound responses in the mPFC are not a focus of the paper, so either this point could be clarified or this aspect could be removed.

The text for Figure 3C states that the example neuron is active for the right spout, but the legend states that it is active for the left spout (which looks to be the case from the figure).

Since prior-related activity is a focus of the paper, it would be helpful to see a time course of activity related to block (as in Figure 3E) rather than just as snapshots in time (Figure 3G). For example, a plot could be provided like Figure 3E, but split according to the conditions in Figure 3F. This would be helpful for other related plots (Figure 4G, Supplementary Figure 6G). Alternatively, if equal fractions of positively and negatively block-modulated cells (Supplementary Figure 4A) average out the effect of the block, it may be useful to split by direction of modulation population, or plot a heatmap of cell activity (like Figure 3B) with the difference between preferred and non-preferred blocks.

Related to this – it is hard to gain an intuition for how the prior is equally encoded (Figure 5C) and decoded (Figure 6B-C) across regions when prior-related activity seems to be much stronger in the mPFC (Figure 3G) and M2 (Supplementary Figure 6G) compared to the AC (Figure 4G). Could the authors comment on this?

ANOVA p-values are missing from line 225 and Supplementary Figure 5A-C.

The example neuron in Figure 4C is used to show tone-selective responses, although there is a very large response for left tones given a right choice (cyan line). Since this leads into the claim that AC activity is primarily sound-driven, it may help to highlight which dot this cell corresponds to on Figure 4D, to give readers an idea of where it fits in this analysis.

Line 255 should presumably reference Figure 4F, since this panel reference was skipped in the text.

Given that the dominant mPFC activity is suggested to be choice-encoding (From Figure 3D) and the dominant AC activity to be sound-encoding (Figure 4D), it is surprising that there is no difference in choice encoding (Figure 5C-D) or decoding (Figure 6C) between the AC and the mPFC. Could the authors address this?

Reviewer #3 (Remarks to the Author):

This study investigates how representations of sensory information in the brain are integrated with prior estimates of action value to compute confidence values for decisions. Several cortical regions are thought to play a role in perceptual and value-based decision-making, including representation of relevant sensory cues, value estimates, and choices, but it is not clear how this information is organized and integrated across distinct cortical circuits. To study this, the authors have developed a perceptual and value-based decision-making task for mice, in which animals learn to discriminate auditory stimuli for reward. In this setting, sensory evidence is manipulated to affect decision confidence, while asymmetric reward payoffs encourage choice bias under uncertainty. The authors find that mice depend more on their value-based bias on low confidence trials, suggesting that they integrate sensory input with prior reward estimates to make decisions. Using high-density electrode recordings targeted to key regions (the medial prefrontal cortex, mPFC; the auditory cortex, AC; and the secondary motor cortex, M2), the authors find that while prior value estimates are distributed across cortical regions, AC and M2 predominantly encode sensory inputs and choice, respectively, and mPFC neurons integrate prior value and sensory inputs to represent confidence-dependent action values.

Overall this is a fairly strong study that extends knowledge of how cortical circuits represent information for complex decision computations. The task design is elegant, and the behavioral data convincingly show that animals use both sensory evidence and learned values to make decisions. The authors collected an extensive set of neurophysiological data spanning several key brain areas, and show that there is interesting regional heterogeneity in how relevant task variables are represented. These conclusions are well supported by several lines of evidence, including single-neuron activity measures and population vector analyses.

The analysis of how different task variables are integrated together, particularly in support of the claim that mPFC neurons combine sensory and prior value information, is a bit weaker. While it is interesting to note that mPFC neurons encode both of these features, there is little evidence shown that they “integrate” them together, or that the representations interact or are related in any way. This conclusion of the paper could be bolstered substantially with additional analysis of the present dataset. In particular, a deeper investigation into the relationship between value and sensory coding within mPFC neurons, and comparison of this with M2 and AC neurons (see major comment 3), would strengthen the conclusion that mPFC neurons play a unique role in integration of sensory and value information.

Minor comments:

- 1) In order to identify task-responsive neurons in each region (mPFC, AC, M2), the authors perform a one-sided statistical test to detect activity increased above baseline in each time bin (Figure 3B). Naively, one might expect representations of task variables to manifest as either increased or decreased activity

relative to baseline, especially in highly recurrently connected areas like the PFC. It would be helpful for the authors to provide some justification for focusing this analysis exclusively on increases in activity.

2) The authors quantify task-responsive neurons at each time point as the fraction of recorded neurons that exceed a p-value cutoff in a signed-rank test (Figures 3B, 4B, 6B). This makes sense to examine the time-course of trial activity within a region. However, because activity in different regions may have distinct characteristics (in terms of noise level, sparsity, presence of oscillatory dynamics, etc), there could be regional differences in false discovery rate. It is not clear how to compare the fraction of task-responsive neurons across regions simply using a constant p-value cutoff. To increase interpretability, the authors could include a shuffle control in these figures to give a measure of the null expectation for fraction of responsive neurons, and compare their observed data with this control.

3) In Figure 5, the authors use GLM regression to analyze single neuron tuning to task variables and compare these across regions. The authors describe in methods that they use 10-fold cross-validation to fit and test the models, but it is not clear in Figure 5 where the cross-validated performance is analyzed. It would be useful to clarify where we are looking at cross-validation metrics to assess overall model performance, and where we are looking at the model comparison analysis based on full GLM fits to quantify tuning to specific task variables (Figure 5C). Additionally, please double check and clarify in methods that the cross-validation procedure splits training/test sets by trial (not by drawing random bins across the session) in order to avoid contamination in the fitting by test data (or highly correlated neighboring timepoints).

4) Figure 5C reports the proportion of neurons in each region that are responsive to each of 3 task variables, and the denominator for this proportion is the number of task-responsive neurons (determined by sign-rank tests on binned activity). Because the fractions of task-responsive neurons are different across regions, the regional comparison of tuning in Figure 5C is difficult to interpret, and could potentially be misleading. In addition to what is shown in Figure 5C, the authors should also report the fraction of sound/choice/value-coding neurons compared to the full population recorded in each region as a means for more direct and fair comparison.

Major comments:

1) Based on the negative correlation between tone index in error vs. correct trials (Figure 3D), the authors conclude that mPFC neurons predominantly encode choice and not tone (which they argue would result in a positive correlation, as in AC data shown in Figure 4D). However, it is not clear from the scatter plot alone whether this correlation is driven by a relatively larger fraction of neurons in the upper left and lower right quadrants (“choice-coding”), or driven by a small number of choice-coding neurons that have large discrimination indices. For example, one alternative possibility is that there is a larger fraction of tone-coding neurons (in upper right and lower left quadrants), but that these neurons tend to

have lower discriminability than choice-coding neurons. It would be informative if the authors could explicitly quantify the fractions of neurons that belong to tone- and choice-coding categories, and show this alongside the scatter plot.

2) Based in part on the observation that block discriminability decreases during tone (Figure 3H, right), and the resemblance of this to the result from their RL model (Figure 2F), the authors argue that mPFC neurons modulate prior values into confidence values. However this negative correlation with time seems to be driven largely or almost entirely by the time point immediately before sound onset (which is not part of the tone epoch). Instead, it looks much more like block coding drops off dramatically at the time of tone onset but does not gradually change during tone (and, in contrast, actually does look like it gradually decreases over time in M2 neurons, Figure 6H). This appears to be the main point of contrast that the authors want to draw between mPFC and M2 neurons, and is therefore relevant to their conclusions about the unique properties of mPFC neurons in this task. It would help to spell this point out a bit more and provide some additional line of evidence showing that the time-course of block coding during the tone is distinct between mPFC and M2 neurons.

3) In general, the authors want to make a claim about the specific role of mPFC neurons, compared with M2 and AC neurons, in integrating sensory input with the prior value estimate to produce confidence values: "The choice and block modulations were increased and decreased, respectively, consistent with the gradual modulation of prior values into confidence values...". If I am understanding correctly, the authors are suggesting that mPFC neurons initially encode prior value (i.e., discriminate preferred block), and then during a given trial these responses evolve in a gradual way to represent the upcoming choice in a manner that is sensitive to the animal's confidence level. However, as shown the data only indicate that block and choice information are both present in the mPFC neuron population, but not that these features are integrated or in some way interact. Apart from the point about the time-course of block coding, the coding features of mPFC and M2 neurons look quite similar. The main claim about the unique features of mPFC neurons could be strengthened substantially.

3a) Is there any evidence that mPFC neurons that code for preferred block before the tone overlap substantially with those that code for choice during late tone? Showing that these two task variables are encoded in the same neurons (or at least overlap in a significant subset of mPFC neurons) would substantiate the claim that there is a gradual modulation of prior values into confidence values in mPFC.

3b) The behavioral evidence that mice modulate prior values into confidence values is related to the difference in their behavior between long tone (presumably high confidence) and short tone (low confidence) conditions. However, the authors focus their analysis of neural data almost exclusively on long tone trials. If mPFC neurons evolve to encode confidence values, one might expect that choice coding is stronger on more confident trials (at the end of long tone compared to short tone trials). The authors can test this, and also compare this feature with M2 neurons, which one might expect to encode choice, but not in a manner that is sensitive to confidence.

Reviewer #4 (Remarks to the Author):

This work studies the decision making in mice in an experimental setup designed to capture the process of integration of sensory evidence with the task structure (prior). With behavioral and neural data analysis, it suggests the capability of mice in integration of sensory evidence with the prior task structure, as well as representation of different task components in different brain regions.

The tackled problem, i.e. combination of perceptual and value-based decision making, is definitely an interesting and necessary focus for the community as the previous studies have been mostly solely perceptual or value-based. The behavioral analysis supports the idea of combination of sensory uncertainty (belief), task structure (prior) and rewards. However, I did find it difficult to convince myself that the neural data from mPFC fully supports this integration. Specifically, it looks like that only the block (more rewarding option), and the choice is encoded in this area. This is different from the combination of belief and reward, which is the result of accumulation of evidence combined by reward. The authors did mention some of these points (such as not seeing accumulated evidence in mPFC) in some parts of the paper. However, I think it does need more clarification, for example in the introduction. Moreover, claims about the confidence-value representation, or combination of belief and value in mPFC (as depicted in Fig S9) are not really supported by the neural analysis results (because they suggest the combination of belief as a probabilistic representation and reward). Overall, the computation of belief for the decision making, which is hallmark of perceptual decision making, is not really supported by the mPFC neural data. Instead, the presentation of block (more rewarding direction) and choice have been observed in mPFC, which is interesting, but probably less impactful result for the community.

This clarification is especially important as it looks like there is mismatch between the behavioral model and the neural data. Specifically, I suspect that the prior-value-only RL model could also explain the neural data, so I think that model should also be used and compared with as the input to the GLM model for mPFC neurons.

I also did not understand why some of the neural data analysis only contain the long duration trials. This could be especially an issue as it looks like the performance of mice in the long duration trials is very high (close to 1). Tasks/trials with intermediate performance/posterior probability better test the “probabilistic computation”. Including results of short-duration trials or an explanation of why they are not included in some analysis would be very much appreciated.

Finally, it looks like that the psychometric function of fig 1 is the pooled data. Please include the psychometric function of each mouse in the supplementary material, especially because the short duration is different among them. Also, it would be great if the authors explain why different short duration is picked for some mice.

We appreciate the reviewers' thoughtful and helpful comments. Our responses to their comments are provided below.

Reviewer #1:

The paper titled "Localized and global computation for integrating prior value and sensory evidence in the mouse cerebral cortex" investigates how the brain integrates prior knowledge of action outcomes and sensory evidence to compute action values. The integration of prior value and sensory inputs in the brain is a broad and fundamental topic for neuroscientists, and the combination of their original behavioral setup, large-scale neuronal recordings in three cortical regions, and sophisticated computational analyses can potentially provide significant insights for understanding this issue. Overall, the study is very interesting and informative, and the results are presented in a clear and structured manner, making them easily understandable. However, I believe that further refinement is needed to improve the clarity of the research objectives, the interpretation of results, and the discussion of limitations. Addressing these points would strengthen the paper and contribute to the existing knowledge in the field of decision-making neuroscience.

We thank the reviewer for the supportive and helpful comments.

Major points:

1) The interpretation of global representations of prior value The major finding of the global representations of prior value is only discussed in a single sentence as below, and the significance of this finding is not clear at all. Elaborating and discussing this, as well as other possibilities and limitations, are crucial. "As the prior values need to be maintained across trials for future actions, such long maintenance may need global coding (lines 416-417)."

In discussion, we added two paragraphs about the global representations of prior (**Page 14, line 488**). We described other possibilities about prior representations such as motor preparation, body movement, or attention as the limitations of this study. We also described that our main findings about prior representations were based on the proportion of neurons based on task responsive neurons (**Figure 5**). Further experiments are essential to investigate whether the prior encoding of early sensory processing is necessary to generate prior-value dependent choices. However, we believe that the existence of prior-modulated neurons across cortical areas was at least provide an important insight for the field of decision making. Similar result is reported on sensory prior in a recent preprint (Findling et al, biorxiv 2023).

1-1) I could not find explicit information regarding the "maintenance of prior." How many trials are required to switch across different blocks? How many past trials are counted for current choice information? Basic information like this is necessary to evaluate the maintenance of prior information for individual animals.

We analyzed the number of trials required to flip the correct behavior performance between the left and right choices at the block changes (**Figure S1b**). We found that the 7 trials were needed to switch the choices.

We also used a logistic regression analysis to verify that the choice behavior was affected by the reward experiences of past multiple trials (**Figure S1c**) (**Page 4, line 123**).

1-2) Reward/punishment is normally represented brain-wide, and what they call the "prior value" may simply be equivalent to sustained reward experience. Are there mPFC neurons that shows sustained activity in response to correct/error outcome to subsequent trials? If there are, what is the relationship between those neurons and the neurons analyzed in Fig.3G?

We found that there were some neurons which showed sustained activity in the mPFC after the correct or error outcome (**Figure S6**). These sustained neurons had almost no overlapping from the group of neurons which activated before sounds (neurons in **Figure 3g**). The activity of sustained neurons had no block modulations before the sound onsets, while the neurons in **Figure 3g** showed the block-modulated activity (**Figure S6**) (**Page 7, line 238**).

1-3) An alternative possibility for this prior representation is movement-related activity. Do the authors record videos of mice behaviors on the treadmill during the task? It is known that task-irrelevant movement significantly influences activities in cortical areas (e.g., Musall et al., Nat. Neuro 2019). The previous reward experience may change their running speed, body orientation, and postural biases, allowing them to optimize their bodies for the following choices. Such biased motor activities might transmit globally to the brain, creating a correlation with prior value and biased choices in subsequent trials.

In 55 out of 97 sessions with neural recording, we captured the facial movement of mice with one camera (**Figure S11c,d**). We used DeepLabCut to capture how the mouse face moved during the trials. We found that the facial movement and prior values were not significantly correlated during task (**Figure S11e,f**).

Next, we used linear regression and analyzed whether the running speeds of mice, detected with rotary encoder, were correlated to the prior reward expectations. We found that the running speeds and prior values did not have significant correlations in some sessions (**Figure S11g,h**). We focused on the sessions with no significant correlations and analyzed whether the neurons represented the prior values (**Figure S11i**). We found that the neurons still encoded the prior values, confirming the prior-value representations (**Page 10, line 334**).

The relationship between the prior representations in our study and the movement, motor preparation, or attention was also described as the limitations of this study in Discussion (**Page 15, line 502**).

2) "Integration" is not directly evaluated

Despite their paper's title, "integration of prior and sensory inputs" are not clearly evaluated at the neural level under their model variables. According to their model, they have trial-by-trial value of $conf_Qa$, which is defined by multiplying the prior value Qa with associated tone category $Prob$. I think that evaluating neuronal activities across areas with these model-based latent variables are one of straightforward way to evaluate their claim and directly comparable with a previous study (Lak et al., 2018).

We directly plotted the neural activity in different levels of prior values and action values (previous conf_Q) in the RL model (**Figure 3i**). Before sound, the neural activity of mPFC was correlated to the prior values than the action values. In contrast, at the end of sound, the activity was correlated to the action values (**Page 8, line 248**).

Linear regression analysis further investigated which task parameters (sound frequency, choice, prior value, combinations of these parameters) were represented in single neurons in addition with the running speed of mice at before, initial, and end of sound (**Figure 3j, 4j, S8j, and Table S1**) (**Methods, Page 28, line 979**). We found that, before sounds, neurons in the AC, mPFC, and M2 represented the prior values. During the sound inputs, the neurons represented the integration of prior values and sound frequencies. The integration of these two parameters were the hallmark of action-value representations (**Page 8, line 254**).

3) Sequential activity

“The confidence-value representations in the mPFC were reported in a previous study¹². We advanced the finding by that a temporal sequence of population neuronal activity in the mPFC modulated the prior-value representations with sensory inputs to compute the confidence values (Figure 3). (line 397)”

“we found that a temporal sequence of neurons in the medial prefrontal cortex of mice updated the prior reward expectations with incoming sensory inputs (lines 23-25)”

“This gradual modulation was represented in a sequential population activity of mPFC neurons (Figure 3). (line 374)”

3-1) *Though they signify the finding of the "temporal sequence" in mPFC, its meaning is not discussed well. How is this sequential recruitment of neurons relevant to the integration of prior and sensory inputs?*

In Discussion, we removed the “temporal sequence” at previous line 397. Instead, we explained in detail how the neurons in mPFC represented the prior values and choices. The neurons, started to increase the activity before sounds, mainly showed the shifts of representations from prior values to choices, while the neurons activated only after sounds were mainly for choice representations (**Figure S7**) (**Page 13, line 456**).

3-2) *They state that they first found that the task-relevant neurons were sequentially recruited during the long-sound trials, rather than mainly observing the ramping activity of neurons toward the end of sounds. However, as they picked up neurons based on the significant increase of neurons at one of the 132 time-windows from baseline, there is no wonder to observe "sequential recruitment" of cells (even random data can create a pseudo sequence). Did the authors cross-validate the sequential recruitment by randomly splitting the trials, particularly the ones activated during the tone presentations?*

We divided the trials in each session into 2 groups and tested whether the sequence in 2 groups were correlated. This cross validation (CV) was repeated 100 times to reduce noise from random grouping. At the same time, we investigated the correlations of shuffled neurons in which the timing of spikes was shuffled but the number of spikes was kept in each neuron. We found that the correlations of temporal sequences in the CV were high ($r = 0.72$ -

0.88 on average in the AC, M2, and mPFC) compared to the correlations in shuffled neurons ($r = 9.4e-5 - 0.0072$). (**Figure S3d, Page 6, line 197**).

3-3) *Since they claim that they found task-relevant neurons typically show peak activity at some point during the tone presentation and not a ramping type, it is more reasonable to show such transient neurons as examples rather than ramping-type neurons in Fig.3C.*

First, we weakened our statement (**Page 6, line 193**): “In addition to observing the ramping-activity of neurons toward the end of sounds, the task-relevant neurons were recruited before and during the long-sound trials.”

We also changed the example neuron to a neuron with transient activity (**Figure 3C**).

3-4) *The authors showed temporal sequence activity in Fig.3B and gradual integration of prior and sensory inputs in Fig.3GH, but there is no direct connection between the temporal sequence and the gradual integration, despite their claim that the temporal sequence updated the prior reward expectations as mentioned above. Do those neurons newly recruited during the sound presentation actually show similar tuning curves as in Fig.3EH?*

Thank you for the advice. We separated the task relevant neurons into two types. (i) One group started to increase the activity before sounds. (ii) The other group increased the activity only after sounds (**Figure S7**). Interestingly, we found that (i) the before-sound neurons mainly increased and decreased the choice modulation and block modulation, respectively. In contrast, (ii) the after-sound neurons mainly increased the choice modulation. The integration of these two neuron groups formed the modulation of value representations in **Figure 3g (Page 7, line 240)**.

4) *Decision confidence*

The use of the term "confidence" is confusing and not justified.

4-1) *First, their use of variable names is not coherent (e.g., action value, Q value, conf_Q value, sensory confidence, confidence). Please make sure they are coherent.*

We agree that the word “confidence” is a probability estimate of correctness (sensory confidence) (e.g., Kepecs et al, nature, 2008), and we should not use “confidence” for the expected reward for each choice (value). At least, we should not mix values and confidence. We re-named the parameter in the RL model. We decided not to use “confidence” in the model. We re-defined the previous conf_Q as action value (e.g., Lak et al, neuron 2020):
action value = P(tone category) x Prior value (**Page 26, line 910**).

4-2) *Second, the use of the term "confidence" is different from that in the previous study (Lak et al., 2018), despite authors utilizing a very similar model. The authors express confidence as the multiplicative reward expectation, whereas Lak et al. describe confidence as a probability estimate of reward (sensory confidence). Utilizing the term "confidence" for their conf_Q value is the authors' original invention and not justified.*

The original paper by Kepecs et al. (Nature 2008) uses confidence as the probability of the “correctness” of the choice given stimulus. Lak et al. follow this convention, showing that their fitted values of "confidence" follow the original signature of decision confidence (such as Fig.5b in Kepecs et al., Nature 2008, Fig.1K in Lak et al., Neuron 2018). Because it is not

obvious that this kind of licking choice paradigm requires the computation of "decision confidence" for a post-decision wagering strategy, it is necessary to confirm such a confidence signature of the previous studies. Alternatively, I would suggest that they stick to objective and general terms such as stimulus difficulty and action value.

Thank you for the advice. Same as the comment on 4-1), we decided not to use "confidence" for the values in our RL model. We re-defined the previous `conf_Q` as action value (Page 26, line 910).

5) Individual variability of animals

The manuscript does not provide much information about the replicability of key findings in terms of different animals.

For instance, Figure 1C shows an example of a session that shows a sigmoid-type PMF, but it is not clear how much individual differences exist in their dataset. It would be helpful to show the psychometric curves with overlaid data from each subject or fitted values from individual animals.

We revised Figure 1c which now showed the average psychometric function of all sessions. We also showed the averaged psychometric function in each mouse. In addition, Figure S1a showed the psychometric function in each session in all the 8 mice (Page 4, line 117).

The core finding of Figure 5C also replicable in at least multiple animals?

In Figure S11a, we plotted the proportion of neurons encoding each task variables in each mouse. Although the results had some variances from the different signal to noise ratio of animals or sessions, the similar tendency was observed as in Figure 5c.

Minor:

1) *I think that the first paragraph of the results section would be more appropriate to be merged into the last paragraph of the introduction.*

We merged the first paragraph of the results section to the last paragraph of the introduction (Page 3, line 73).

2) *"These studies suggest a hypothesis that the cerebral cortex, including the sensory and frontal cortices, are orchestrated to integrate prior knowledge and sensory stimuli, although the role of each cortical region is still unclear." (lines 59-61)*

It is stated that "the role of each cortical region is unclear," despite there being multiple previous studies that show the combined representation of prior and sensory inputs in different brain regions.

"A recent study reported that the mPFC represents a confidence value that combines prior values and sensory inputs, although it is unclear where and how prior values and sensory inputs are represented and integrated in the brain." (lines 54-55)

This sentence is also repetitive. While the authors state that Lak et al. showed "confidence value combines prior value and sensory inputs in mPFC," the latter half of the sentence still states "it is unclear where and how." Authors should elaborate on what is missing in the previous studies.

Thank you for organizing the Introduction. We revised the manuscript and introduced what is missing in the previous studies. We emphasized that previous studies mainly focused on specific brain areas in different tasks, which makes it difficult to observe how the cerebral cortical areas are organized to optimize choices (**Page 2, line 62**).

3) Figure 3: The authors described that they removed neurons recorded from PPC in the results section. However, since the reasoning is only given qualitatively, the necessity to remove the data from the analysis is not convincing. Given that this paper claims that prior information is represented "globally" across cortical areas, it is curious to know whether this finding is generalizable to PPC.

Adding PPC analysis only for the Fig.3 encoding analysis will further support their claim (no need to include it in the paper if they are not comfortable). Alternatively, please justify why they could not include the PPC data in a quantitative way.

The number of neurons recorded from the PPC was 55 on average per session which were less than half of neurons recorded from the mPFC, M2, and AC (Mann-Whitney U-test, $p = 2.3e-9 - 1.4e-4$) (**Page 5, line 170**). Also, the number of task-relevant neurons in PPC were 22 on average which was less than one third of those from the other three regions ($p = 3.5e-7 - 1.7e-6$) (**Methods, Page 27, line 953**). Thus, we decided not to include PPC for the analysis.

4) Figure 5C:

... Thus, the sounds and choices were encoded in the AC and M2, respectively, while the prior values were encoded in all 3 recorded regions. (lines 293-294)

This statement is not accurate for describing their results. What they found is that the relative proportion of neurons encodings for sounds and choices differs across areas, and there are no significant differences in the encodings of prior values among the three areas (I mean it is not like "X is encoded in A and Y is encoded in B").

Thank you. We revised the manuscripts as you advised (**Page 10, line 328**).

Reviewer #2:

This paper investigates how cortical activity is related to stimuli, choices, and reward value in the auditory (AC), secondary motor (M2), and medial prefrontal (mPFC) cortex. The authors were able to examine the interaction between these features by combining an auditory sensory discrimination task with perceptual uncertainty and block-based reward biases. Behaviorally, they demonstrate that animals combine stimulus information with reward block history to produce value-based choices. From their neural recordings, they show that sound-related activity is predominantly in the AC, choice-related activity is predominantly in M2 and mPFC, and all three areas contain activity related to reward bias block. While the manuscript would benefit from certain clarifications, it provides interesting insight that block-related activity exists in all three areas, and that this activity is affected by choice in M2 but not in AC or mPFC.

We thank the reviewer for the supportive and helpful comments.

Specific comments:

(1) In Figure 1E, “former” and “latter” are not defined, and the methods section references another paper for further explanation. Please define this in the figure legend, and include at least a summary of methods which are more fully explained in previous papers.

In **Figure 1e**, we revised the “former” and “latter” as “former half of tone” and “latter half of sound” to give a clear explanation. We added the detailed explanation of analysis in the **legend of Figure 1e (Page 16, line 554)** and in **Methods (Page 25, line 869)**. In Methods, we also added two references (Katz et al, nature, 2016; Yates et al, nat. neurosci, 2017) to explain the analysis.

(2) In Figure 3A, this is presumably only long sound trial data, though this isn't stated. Please state this in the figure legend.

Yes, the figures are only from the long-sound trials. We added the sentence in the **legend of Figure 3a** that the data were only from the long-sound trials (**Page 17, line 593**).

(3) In Figure 3B and related text, it's unclear whether there is a true sequence of activity spanning the sound in the mPFC, as opposed to a group of cells which are generally preferentially active during that period. One graphical aspect of this is the yellow stripe through Figure 3B (and other heatmaps) – this is present by definition since activity is z-scored and sorted, the authors might try sorting by one half of the trials and plotting the other half of trials to provide a clearer idea of how robust the sequence is.

We divided the trials in each session into 2 groups and tested whether the sequence in 2 groups were correlated. This cross validation (CV) was repeated 100 times to reduce noise from random grouping. At the same time, we investigated the correlations of shuffled neurons in which the timing of spikes was shuffled but the number of spikes was kept in each neuron. We found that the correlations of temporal sequences in the CV were high ($r = 0.72 - 0.88$ on average in the AC, M2, and mPFC) compared to the correlations in shuffled neurons ($r = 9.4e-5 - 0.0072$). (**Figure S3d, Page 6, line 197**).

*The Δ activated neurons plot also shows a bump at time 0, which could indicate a small fraction of consistently active cells from tone onset rather than a ramp. The authors also state in line 252 that AC activity did not ramp “in contrast with the mPFC” (line 252 – possibly referring to the ramping of the orange line in Fig 3E?), making it less clear whether there is a ramp or not. This is all a minor point since the sound responses in the mPFC are not a focus of the paper, so either this point could be clarified or this aspect could be removed. We agree that it is difficult to quantify or to define what is ramping of neurons. We therefore removed the sentence comparing the ramping of neurons (**Page 9, line 284**).*

*(4) The text for Figure 3C states that the example neuron is active for the right spout, but the legend states that it is active for the left spout (which looks to be the case from the figure). We revised the caption of **Figure 3c** which now has a different neuron based on a comment from another reviewer (**Figure 3c and caption (Page 17, line 601)**).*

*(5) Since prior-related activity is a focus of the paper, it would be helpful to see a time course of activity related to block (as in Figure 3E) rather than just as snapshots in time (Figure 3G). For example, a plot could be provided like Figure 3E, but split according to the conditions in Figure 3F. This would be helpful for other related plots (Figure 4G, Supplementary Figure 6G). Alternatively, if equal fractions of positively and negatively block-modulated cells (Supplementary Figure 4A) average out the effect of the block, it may be useful to split by direction of modulation population, or plot a heatmap of cell activity (like Figure 3B) with the difference between preferred and non-preferred blocks. Thank you for the advice. We plotted the average activity during the preferred and non-preferred blocks and plotted how the distributions of positive and negative modulations of neurons were changed by time (**Figure 3f, 4f, S8f**). We also made a heatmap of how the activity of single neurons was modulated by blocks (**Figure S5c (Page 7, line 234)**).*

(6) Related to this – it is hard to gain an intuition for how the prior is equally encoded (Figure 5C) and decoded (Figure 6B-C) across regions when prior-related activity seems to be much stronger in the mPFC (Figure 3G) and M2 (Supplementary Figure 6G) compared to the AC (Figure 4G). Could the authors comment on this?

For encoding, the GLM analysis detected how many neurons represented each task variable, while the plotting of activity showed the magnitude of modulation. For example, in the AC, the block-dependent changes of activity were positively or negatively observed compared to the tone-frequency preference. Thus, in the population level, it is hard to see the magnitude difference in the AC (**Figure 4g**). But there were similar number of neurons encoding the prior values in the AC as well as in the M2 and mPFC (**Figure 5c and S5c**).

For decoding, previous studies show that only a handful of neurons or even a single neuron can decode a task parameter as well as animal behavior (Britten et al, J Neurosci, 1992; Heuer et al, J Neurophysiol, 2004; Christison-Lagay et al, J Neurophysiol, 2017). Indeed, in the AC, there were some neurons strongly encoding the prior values as well as the M2 or mPFC (**Figure 5d**). These neurons potentially provided the performance of prior-value decoding in the AC. We added these detail comments on Discussion (**Page 14, line 482**).

(7) ANOVA p-values are missing from line 225 and Supplementary Figure 5A-C.

We added the p-values of ANOVA within the panels of figures (not in legends). In the figure legend, we explained that the p-values of ANOVA are shown as the figure insets (**Figure S7; Supplementary line 144** (previous Figure S5)).

(8) The example neuron in Figure 4C is used to show tone-selective responses, although there is a very large response for left tones given a right choice (cyan line). Since this leads into the claim that AC activity is primarily sound-driven, it may help to highlight which dot this cell corresponds to on Figure 4D, to give readers an idea of where it fits in this analysis. In **Figure 4d**, we highlighted the example AC neuron in **Figure 4c**. We also highlighted the example neurons in the mPFC and M2 in **Figure 3d and S8d**, respectively.

(9) Line 255 should presumably reference Figure 4F, since this panel reference was skipped in the text.

Thank you. We added the reference to **Figure 4e** (previous Figure 4F) on **Page 8, line 286**.

(10) Given that the dominant mPFC activity is suggested to be choice-encoding (From Figure 3D) and the dominant AC activity to be sound-encoding (Figure 4D), it is surprising that there is no difference in choice encoding (Figure 5C-D) or decoding (Figure 6C) between the AC and the mPFC. Could the authors address this?

Neurons in the auditory cortex mainly represented the tone frequency but the neural activity was also modulated by choices even during sound (**Figure S10, Page 8, line 281**). This in part may contribute to the choice encoding and decoding in the auditory cortex (**Discussion, Page 14, line 471**).

Reviewer #3:

This study investigates how representations of sensory information in the brain are integrated with prior estimates of action value to compute confidence values for decisions. Several cortical regions are thought to play a role in perceptual and value-based decision-making, including representation of relevant sensory cues, value estimates, and choices, but it is not clear how this information is organized and integrated across distinct cortical circuits. To study this, the authors have developed a perceptual and value-based decision-making task for mice, in which animals learn to discriminate auditory stimuli for reward. In this setting, sensory evidence is manipulated to affect decision confidence, while asymmetric reward payoffs encourage choice bias under uncertainty. The authors find that mice depend more on their value-based bias on low confidence trials, suggesting that they integrate sensory input with prior reward estimates to make decisions. Using high-density electrode recordings targeted to key regions (the medial prefrontal cortex, mPFC; the auditory cortex, AC; and the secondary motor cortex, M2), the authors find that while prior value estimates are distributed across cortical regions, AC and M2 predominantly encode sensory inputs and choice, respectively, and mPFC neurons integrate prior value and sensory inputs to represent confidence-dependent action values.

Overall this is a fairly strong study that extends knowledge of how cortical circuits represent information for complex decision computations. The task design is elegant, and the behavioral data convincingly show that animals use both sensory evidence and learned values to make decisions. The authors collected an extensive set of neurophysiological data spanning several key brain areas, and show that there is interesting regional heterogeneity in how relevant task variables are represented. These conclusions are well supported by several lines of evidence, including single-neuron activity measures and population vector analyses.

The analysis of how different task variables are integrated together, particularly in support of the claim that mPFC neurons combine sensory and prior value information, is a bit weaker. While it is interesting to note that mPFC neurons encode both of these features, there is little evidence shown that they “integrate” them together, or that the representations interact or are related in any way. This conclusion of the paper could be bolstered substantially with additional analysis of the present dataset. In particular, a deeper investigation into the relationship between value and sensory coding within mPFC neurons, and comparison of this with M2 and AC neurons (see major comment 3), would strengthen the conclusion that mPFC neurons play a unique role in integration of sensory and value information.

We thank the reviewer for the supportive and helpful comments.

Minor comments:

1) In order to identify task-responsive neurons in each region (mPFC, AC, M2), the authors perform a one-sided statistical test to detect activity increased above baseline in each time bin (Figure 3B). Naively, one might expect representations of task variables to manifest as either increased or decreased activity relative to baseline, especially in highly recurrently

connected areas like the PFC. It would be helpful for the authors to provide some justification for focusing this analysis exclusively on increases in activity.

Thank you for the advice. In the mPFC, previous study shows that the neurons with decreasing the activity during task were less modulated by task variables than the neurons with increasing the activity (Le Merre et al, neuron, 2018). In other areas (e.g., auditory cortex or hippocampus), the increasing neurons were selectively analyzed to investigate the sensory representations or task related activity (MacDonald et al, neuron, 2011; Thomas et al, J Neurosci, 2020). We thus decided to analyze the increasing neurons as the uniform criteria across regions (**Page 6, line 186**).

We also verified that the number of decreasing neurons during sounds in our task was at least 10 times smaller than the increasing neurons in the mPFC and M2 (**Figure S4 and S5c**). In the auditory cortex (AC), the distribution of block modulated activity in the decreasing neurons were significantly smaller than that in the increasing neurons (**Figure S4**).

2) The authors quantify task-responsive neurons at each time point as the fraction of recorded neurons that exceed a p-value cutoff in a signed-rank test (Figures 3B, 4B, 6B). This makes sense to examine the time-course of trial activity within a region. However, because activity in different regions may have distinct characteristics (in terms of noise level, sparsity, presence of oscillatory dynamics, etc), there could be regional differences in false discovery rate. It is not clear how to compare the fraction of task-responsive neurons across regions simply using a constant p-value cutoff. To increase interpretability, the authors could include a shuffle control in these figures to give a measure of the null expectation for fraction of responsive neurons, and compare their observed data with this control.

Based on the distribution of baseline neural activity (as null distribution), we analyzed the relationship between the p-values of the Wilcoxon signed-rank test (for task-relevant neurons) and those based on the null distribution (**Figure S3a-c**). The definition of baseline activity was identical to that for the task relevant neurons (i.e., inter-trial-interval for sound-aligned activity and before choice for choice-aligned activity; **Methods**). We found that there were strong correlations between the two p-values in all the recorded areas (Spearman correlation $r = 0.54 - 0.65$). Also, the proportion of neurons detected as task-relevant was 45%, 61%, 32% in the AC, M2, and mPFC with the Wilcoxon signed-rank test ($p < 1e-10$). The order of proportions among 3 recorded areas was identical with the criteria based on the null distribution ($p < 0.01$) (8.2%, 20%, 6.9% in the AC, M2, and mPFC).

3) In Figure 5, the authors use GLM regression to analyze single neuron tuning to task variables and compare these across regions. The authors describe in methods that they use 10-fold cross-validation to fit and test the models, but it is not clear in Figure 5 where the cross-validated performance is analyzed. It would be useful to clarify where we are looking at cross-validation metrics to assess overall model performance, and where we are looking at the model comparison analysis based on full GLM fits to quantify tuning to specific task variables (Figure 5C). Additionally, please double check and clarify in methods that the cross-validation procedure splits training/test sets by trial (not by drawing random bins across the session) in order to avoid contamination in the fitting by test data (or highly correlated neighboring timepoints).

We used 10-fold cross validation to determine the elastic net regularization λ using a software package of cvglmnet in MATLAB (<https://glmnet.stanford.edu/index.html>). The cross validation split the data by trials and not by randomly splitting the time points. To make sure this, we re-analyzed **Figure 5**, but the overall results did not change. With the estimated λ , we analyzed the regression coefficients of GLM using all the data. We compared the model fitting between the full model and models without task parameters to identify which parameters were represented in single neurons (**Figure 5**). We revised the main text (**Page 9, line 308**) and methods (**Page 29, line 1010**) to clarify our GLM.

4) Figure 5C reports the proportion of neurons in each region that are responsive to each of 3 task variables, and the denominator for this proportion is the number of task-responsive neurons (determined by sign-rank tests on binned activity). Because the fractions of task-responsive neurons are different across regions, the regional comparison of tuning in Figure 5C is difficult to interpret, and could potentially be misleading. In addition to what is shown in Figure 5C, the authors should also report the fraction of sound/choice/value-coding neurons compared to the full population recorded in each region as a means for more direct and fair comparison.

We plotted the proportion of neurons detected by GLM analysis among all the neurons recorded (**Figure S11b**). Because the proportion of task relevant neurons in the mPFC was smaller than that in the AC and M2, the proportion of neurons encoding prior values became small in the mPFC.

Major comments:

1) Based on the negative correlation between tone index in error vs. correct trials (Figure 3D), the authors conclude that mPFC neurons predominantly encode choice and not tone (which they argue would result in a positive correlation, as in AC data shown in Figure 4D). However, it is not clear from the scatter plot alone whether this correlation is driven by a relatively larger fraction of neurons in the upper left and lower right quadrants (“choice-coding”), or driven by a small number of choice-coding neurons that have large discrimination indices. For example, one alternative possibility is that there is a larger fraction of tone-coding neurons (in upper right and lower left quadrants), but that these neurons tend to have lower discriminability than choice-coding neurons. It would be informative if the authors could explicitly quantify the fractions of neurons that belong to tone- and choice-coding categories, and show this alongside the scatter plot.

We added a decoding analysis which investigated the area under the receiver operating curve (auROC) for separating the neural activity of choice or sound categories (**Figure 3d, 4d, S8d**) (**Page 6, line 208**) (**Methods, Page 28, line 963**). Consistent with the tone-index analyses, neurons in the mPFC and M2 had better discrimination of choice than sound, while the AC had a good sound decoding. We also added the auROC analysis for decoding prior values in single neurons (**Figure S5b**).

2) Based in part on the observation that block discriminability decreases during tone (Figure 3H, right), and the resemblance of this to the result from their RL model (Figure 2F), the

authors argue that mPFC neurons modulate prior values into confidence values. However this negative correlation with time seems to be driven largely or almost entirely by the time point immediately before sound onset (which is not part of the tone epoch). Instead, it looks much more like block coding drops off dramatically at the time of tone onset but does not gradually change during tone (and, in contrast, actually does look like it gradually decreases over time in M2 neurons, Figure 6H). This appears to be the main point of contrast that the authors want to draw between mPFC and M2 neurons, and is therefore relevant to their conclusions about the unique properties of mPFC neurons in this task. It would help to spell this point out a bit more and provide some additional line of evidence showing that the time-course of block coding during the tone is distinct between mPFC and M2 neurons.

*Thank you for checking our data in detail. **Figure S9** made a comparison of choice and block modulations between the mPFC and M2 (**Page 8, line 264**).*

We used 10-fold cross validation (CV) to investigate whether the neural activity of mPFC and M2 fit to 2 independent robust linear regressions (for mPFC and M2) or 1 regression by comparing the residual sum of squared error (RSS) of two models.

We found that the choice modulations of mPFC and M2 were fit to 2 different regressions rather than 1 regression, suggesting that the M2 had larger choice modulations than the mPFC. The prior modulations of mPFC and M2 also fit to 2 regressions. However, as you pointed, when we did not include the activity before sounds, 1 robust linear regression was enough to explain the prior modulations of two regions. These results suggest that the prior modulation was slightly larger in the mPFC than M2, and the difference of block modulations mainly came from the activity before sound.

3) In general, the authors want to make a claim about the specific role of mPFC neurons, compared with M2 and AC neurons, in integrating sensory input with the prior value estimate to produce confidence values: “The choice and block modulations were increased and decreased, respectively, consistent with the gradual modulation of prior values into confidence values...”. If I am understanding correctly, the authors are suggesting that mPFC neurons initially encode prior value (i.e., discriminate preferred block), and then during a given trial these responses evolve in a gradual way to represent the upcoming choice in a manner that is sensitive to the animal’s confidence level. However, as shown the data only indicate that block and choice information are both present in the mPFC neuron population, but not that these features are integrated or in some way interact. Apart from the point about the time-course of block coding, the coding features of mPFC and M2 neurons look quite similar. The main claim about the unique features of mPFC neurons could be strengthened substantially.

3a) Is there any evidence that mPFC neurons that code for preferred block before the tone overlap substantially with those that code for choice during late tone? Showing that these two task variables are encoded in the same neurons (or at least overlap in a significant subset of mPFC neurons) would substantiate the claim that there is a gradual modulation of prior values into confidence values in mPFC.

Thank you for the advice. We separated the neurons into two types. (i) One group started to increase the activity before sound. (ii) The other group of neurons increased the activity only after sound onsets (**Figure S7**). We found that (i) the neurons increased the activity before sound had a similar activity pattern as in **Figure 3g**. The neurons mainly increased and decreased the choice and block modulations during sounds, respectively, suggesting that the change of modulations was encoded in the same neurons (**Page 7, line 240**).

3b) The behavioral evidence that mice modulate prior values into confidence values is related to the difference in their behavior between long tone (presumably high confidence) and short tone (low confidence) conditions. However, the authors focus their analysis of neural data almost exclusively on long tone trials. If mPFC neurons evolve to encode confidence values, one might expect that choice coding is stronger on more confident trials (at the end of long tone compared to short tone trials). The authors can test this, and also compare this feature with M2 neurons, which one might expect to encode choice, but not in a manner that is sensitive to confidence.

We compared the performance of choice decoding between the short- and long-sound trials and among the different tone difficulties. Both in the mPFC and M2, we found that the choice decoding was better in long than in short sound trials (**Figure 6c**), and better in easy- than in difficult-sound trials (**Figure S13c**), suggesting that the choice decoding was stronger in confident trials.

At the choice timing, the decoding performance was similar between the long- and short-sound trials in the M2, while, in the mPFC, the performance was slightly but significantly better in the long- than in the short-sound trials (**Figure S13c**). Also, the M2 neurons had better choice decoding than the mPFC (**Figure 6d**). These results suggest that the M2 neurons strongly represented the choices (**Page 11, line 367; Page 13, line 461**).

Reviewer #4:

This work studies the decision making in mice in an experimental setup designed to capture the process of integration of sensory evidence with the task structure (prior). With behavioral and neural data analysis, it suggests the capability of mice in integration of sensory evidence with the prior task structure, as well as representation of different task components in different brain regions.

We thank the reviewer for the supportive and helpful comments.

(1) The tackled problem, i.e. combination of perceptual and value-based decision making, is definitely an interesting and necessary focus for the community as the previous studies have been mostly solely perceptual or value-based. The behavioral analysis supports the idea of combination of sensory uncertainty (belief), task structure (prior) and rewards. However, I did find it difficult to convince myself that the neural data from mPFC fully supports this integration. Specifically, it looks like that only the block (more rewarding option), and the choice is encoded in this area. This is different from the combination of belief and reward, which is the result of accumulation of evidence combined by reward. The authors did mention some of these points (such as not seeing accumulated evidence in mPFC) in some parts of the paper. However, I think it does need more clarification, for example in the introduction.

Thank you for the comments. We revised the introduction to clarify that we investigated how the neurons in various cortical areas integrate the reward expectation and the belief of context which is based on the accumulation of sensory inputs (**Page 2, line 36, 40, and 43**).

Moreover, claims about the confidence-value representation, or combination of belief and value in mPFC (as depicted in Fig S9) are not really supported by the neural analysis results (because they suggest the combination of belief as a probabilistic representation and reward). Overall, the computation of belief for the decision making, which is hallmark of perceptual decision making, is not really supported by the mPFC neural data. Instead, the presentation of block (more rewarding direction) and choice have been observed in mPFC, which is interesting, but probably less impactful result for the community.

We directly compared the neural activity with the parameters in reinforcement learning model, i.e., prior values and action values (previous confidence values) (**Figure 3i, 4i, S8i**). We found that, before sounds, the activity was correlated to the prior values than the action values in the mPFC and M2. In contrast, during sounds, the activity was correlated to the action values, supporting (i) the modulation of neural activity and (ii) the action-value representations in the mPFC (**Page 8, line 248**).

(2) This clarification is especially important as it looks like there is mismatch between the behavioral model and the neural data. Specifically, I suspect that the prior-value-only RL model could also explain the neural data, so I think that model should also be used and compared with as the input to the GLM model for mPFC neurons.

We made a regression analysis to investigate whether the regression model with only prior values could explain the neural data in different time windows (**Figure 3j, 4j, S8j; Methods**

Page 28, line 979). As you pointed, before the sound onset, the model with only the prior values fit the neural activity of AC, mPFC, and M2. In contrast, at the initial and end of sounds, the regression model with prior value and sound frequency fit the neural activity (**Figure 3j, Table S1**). The integration of prior value and sound inputs were the hallmark of action-value computation, suggesting the action-value representations in the mPFC (**Page 8, line 254**).

(3) I also did not understand why some of the neural data analysis only contain the long duration trials. This could be especially an issue as it looks like the performance of mice in the long duration trials is very high (close to 1). Tasks/trials with intermediate performance/posterior probability better test the “probabilistic computation”. Including results of short-duration trials or an explanation of why they are not included in some analysis would be very much appreciated.

Thank you for the advice. We compared the performance of choice decoding between short- and long-sound trials and among different tone difficulties (**Figure 6c and S13c**). In the mPFC, we found that the choice decoding was better in long than in short sound trials (**Figure 6c**), and better in easy- than in difficult-sound trials (**Figure S13c**). These results suggest that the choice decoding was stronger in confident trials. We also found a similar tendency in the M2.

As the performance of choice decoding was better in the M2 than in the mPFC (**Figure 6d**), these results suggest that both the M2 and mPFC neurons modulated the activity by sensory confidence, but the M2 neurons had strong choice modulations (**Page 11, line 367; Page 13, line 461**).

(4) Finally, it looks like that the psychometric function of fig 1 is the pooled data. Please include the psychometric function of each mouse in the supplementary material, especially because the short duration is different among them. Also, it would be great if the authors explain why different short duration is picked for some mice.

We revised **Figure 1c** which now showed the average psychometric function of all sessions. We also showed the averaged psychometric function in each mouse. In addition, **Figure S1a** showed the psychometric function in each session in all the 8 mice (**Page 4, line 117**).

During the training phase of behavioral task, we gradually shorten the duration of short sounds. We found that, in some mice, the sound duration of 0.2 s was difficult to make the accurate tone-frequency dependent choices. For these mice, we set the duration of short sounds as 0.4 s. We explained this on Methods (**Page 23, line 792**).

REVIEWER COMMENTS

Reviewer #1 (Remarks to the Author):

The authors have submitted a revised manuscript that largely addresses my concerns with the original submission. I appreciate the effort the authors took to address my comments.

I have a few points still require further clarification.

As evident from the title and abstract, a central theme of the manuscript revolves around the "integration" of prior value and sensory evidence. In their previous paper, the authors demonstrated gradual modulations of the prior value signal by sensory evidence information, at the level of average neural activities of subpopulations, particularly in mPFC. However, in the systems neuroscience field, "Integration" usually means "non-additive" modulations. Thus, in my previous comment, I asked the authors to evaluate them as their model-based conf_Qa (now "action value"), which is the multiplicative of prior and sensory evidence.

Now, they showed average tuning of the action value in their new Fig.3i, which indicates that mPFC holds both prior and sensory evidence information on average, but it does not directly demonstrate that individual neurons integrate them. To demonstrate that, they need to show the tuning of individual neurons for the action value. However, instead, they added Fig.3j, which rather showed the tuning of individual neurons for "prior + sound" (additive model).

Hence, my concern remains that the authors have not yet provided compelling evidence to demonstrate that mPFC neurons genuinely "integrate" this information at the individual neuron level. To assert that mPFC neurons integrate prior and sensory evidence, the authors could show that their action value model fits better than the additive model for each neuron. Alternatively (if individual neurons are too noisy), they could compare average tuning curve with those from additive and multiplicative models.

In response to my comment, the authors also added individual mouse data of psychometric functions (Fig.S1), but as far as I see, individual PMFs appear to be rather explained as the additive model (for instance, see Cohen et al., 2023 Neuron. <https://pubmed.ncbi.nlm.nih.gov/37295419/>). The distinction between additive and multiplicative operations is pivotal, as it's possible that animals can perform their tasks by dynamically focusing on either value or sensory evidence on a trial-by-trial basis, which could potentially yield similar psychometric functions in Figure S1. If that is indeed the case, it wouldn't be surprising if there are no multiplicative operations at the neuronal level.

While I acknowledge that this issue does not impact the authors' main conclusion regarding the global and local representations of priors, I would suggest that the term "integration" be handled with caution. Considering terms like "additive combination" might be more appropriate, but I'm open to seeing evidence that supports the use of "integration" if the authors disagree with this suggestion.

Minor)

The authors added a logistic regression analysis that illustrates the effects of sounds and choice-outcome associations in past trials (Fig. S1c). It shows that the absolute coefficients for the sounds presented in past trials appear to be even larger than those for choice-outcomes.

1) It doesn't make sense to me that the sign of the coefficients becomes significantly reversed (negative) for the sounds in past trials. This suggests that animals tend to choose the left side when the previous sound was associated with a rightward choice in a tone cloud? If that is the case, is there any adaptive reasons that they take such a strategy in this particular task? Could you confirm if my understanding is correct?

2) Does this indicate that the previous sounds have a stronger influence on choices than, or at least an influence almost equal to, past rewarded choices? Is it possible that the prior representations associated with the past reward is contaminated with the stimulus (or associated its choice side) history? Does this affect on their discovery of the past reward representations in AC?

Reviewer #2 (Remarks to the Author):

The revised paper has improved on the original and largely addressed my previous comments. The following points could benefit from further clarification.

1. It remains unclear whether there is a "sequence" of activity after sound onset, and certain aspects of Supplementary Figure 3d are unclear. Since this is a minor point in the scope of the paper, it might be worth clarifying the meaning or removing emphasis on the sequence. Specifically:

This may be an issue of definitions: "sequence" can imply that there are middle-preferring neurons (e.g. neuron A, then B, then C). However, the heatmaps and shaded regions in Supplementary Figure 3d (right) look more like there are two groups: one which is active shortly after sound onset, and another which ramps up near the end of the sound. If this is indeed what the authors mean by "sequence" based

on the context in lines 193-198, it would be helpful to explicitly state this. If not, it remains unclear whether there are middle steps in the sequence, since the spike shuffling method would only test against randomly distributed activity.

Which neurons are plotted in Supplementary Figure 3d? The text and legends seem to suggest that it is all task-relevant neurons as in Figure 3d, but the supplementary figure appears to be showing a subset of task-relevant neurons. Please clarify which neurons are being plotted, and how they are defined if it is a subset.

How is the data in Supplementary Figure 3d normalized? It doesn't appear to be range-normalized as in Figure 3b. Please clarify this in the legend.

The red lines in Supplementary Figure 3d make it difficult to see whether there are underlying sequences, ideally they would be removed.

It is unclear how correlation was done between the split groups of trials: was the activity of all cells concatenated, or were correlations made separately for each cell and then combined?

2. The new plots of activity by block in Figure 3f, 4f, and S8f are helpful, though it appears that the plots of activity by tone have dramatically changed from the original paper (e.g. Figure 3f (left) in the revised paper, compared to Figure 3e in the original paper). It is unclear what would have caused this change from the text and legends, could the authors clarify?

3. Typos:

Figure 3f legend: "starndard"

Line 236: "ANOVA p value" – presumably this meant to include a p-value? Or if it is meant just to identify the test, it could read "x-way ANOVA"

Reviewer #3 (Remarks to the Author):

The authors have done a very nice job with this revision, and have comprehensively addressed my comments with several additional analyses and textual clarifications. The additional paragraphs in the discussion are helpful and provide important context for interpreting the unique contributions of this

study. My only additional comment is just to ask the authors to clean up the discussion paragraphs a bit, as it appears there are quite a few typos there.

With these small edits, I am happy to recommend the manuscript for publication, and would like to thank the authors for their commendable effort in carrying out these studies and revisions.

Reviewer #4 (Remarks to the Author):

I am overall happy with the study and the extensive analysis. However, a few changes is still necessary to me.

1) The authors changed terms related to “confidence-value” to “action value”. First of all, I really appreciate removal of “confidence-value” terms. As one the other reviewers also noted that term was confusing. The new term, however, is still a little bit confusing (with “choice”). “Expected value” or “Expected value of action” as in reference [13] are better choices to me. Alternatively, the authors could clarify the meaning of “action value” from the beginning (and also more explicitly). Currently, “action-value” is defined relatively early in the paper (line 75), but I think the term “action-value” has been used even before line 75.

2) I am still not convinced that mPFC decodes expected value/action value as opposed to choice. The authors has acknowledged this in a few places in the paper, but looks like the main narrative is still about the mPFC mainly decodes the expected value while M2 mainly decodes choices. The fact that expected value has been observed in all areas later in the trial should be clearly stated from the beginning, and small differences between areas and speculations should follow after that. Notably, given the strong correlation between choice and expected value in this study, I do not even expect to see significant difference in decodability in any region.

3) Most importantly, I do not think analysis that produces Table S1, and panel j of figures 3, 4, and S8 is correct at all. There is no statistical test in that analysis, and as the numbers show there is a huge variance in results. Moreover, “choice” is highly correlated with “prior + sound” so they can not be compared as features of a regression-based analysis. This analysis should be removed. Interestingly, results of this analysis (Table S1) also contradict with claims of the paper that M2 is mainly about choice, and AC is mainly about the sound.

We appreciate the reviewers' thoughtful and helpful comments. Our responses to their comments are provided below.

Reviewer #1:

The authors have submitted a revised manuscript that largely addresses my concerns with the original submission. I appreciate the effort the authors took to address my comments.

I have a few points still require further clarification.

We thank the reviewer for the supportive and helpful comments.

Major points:

As evident from the title and abstract, a central theme of the manuscript revolves around the "integration" of prior value and sensory evidence. In their previous paper, the authors demonstrated gradual modulations of the prior value signal by sensory evidence information, at the level of average neural activities of subpopulations, particularly in mPFC. However, in the systems neuroscience field, "Integration" usually means "non-additive" modulations. Thus, in my previous comment, I asked the authors to evaluate them as their model-based $\text{conf_}Qa$ (now "action value"), which is the multiplicative of prior and sensory evidence.

Now, they showed average tuning of the action value in their new Fig.3i, which indicates that mPFC holds both prior and sensory evidence information on average, but it does not directly demonstrate that individual neurons integrate them. To demonstrate that, they need to show the tuning of individual neurons for the action value. However, instead, they added Fig.3j, which rather showed the tuning of individual neurons for "prior + sound" (additive model).

Hence, my concern remains that the authors have not yet provided compelling evidence to demonstrate that mPFC neurons genuinely "integrate" this information at the individual neuron level. To assert that mPFC neurons integrate prior and sensory evidence, the authors could show that their action value model fits better than the additive model for each neuron. Alternatively (if individual neurons are too noisy), they could compare average tuning curve with those from additive and multiplicative models.

Thank you for the comment. As you pointed, the action value is the multiplication of sensory evidence and prior values, but we only tested the correlations between the neural activity and additive model of prior value and sensory evidence in our previous manuscript.

We first checked whether the activity of mPFC neurons correlated to the multiplicative model (action value) rather than the additive model of prior value and sound (**Figure 3i and S8a**).

We found that, during sound, there were no significant differences in the correlations between the multiplicative and additive models (Wilcoxon signed rank test, $p = 0.052$) (**Figure S8a; Page 8, line 254**).

We also analyzed whether the neural activity of mPFC was fit to the linear regression with the additive or multiplicative model (**Figure S8b**). We again found that there was no significant difference in the model fitting (Wilcoxon signed rank test, $p = 0.065$).

These results show that, as you expected, there was no clear evidence that the mPFC neurons represented the action values. The neurons additively represented the prior values and choices during sound (**Page 8, line 259**).

In response to my comment, the authors also added individual mouse data of psychometric functions (Fig.S1), but as far as I see, individual PMFs appear to be rather explained as the additive model (for instance, see Cohen et al., 2023 Neuron.

<https://pubmed.ncbi.nlm.nih.gov/37295419/>).

In addition to our RL model which used the action value (multiplicative RL model) (**Figure 2**), we fit the mouse behavior with an additive RL model which used the ‘prior value + sound evidence’ for value updating. The average Bayesian Information criterion (BIC) per mouse of multiplicative RL model (521.883) was smaller than that of additive RL model (523.233), suggesting that the multiplicative RL model fit to the mouse choices (**Page 26, line 908**).

The distinction between additive and multiplicative operations is pivotal, as it's possible that animals can perform their tasks by dynamically focusing on either value or sensory evidence on a trial-by-trial basis, which could potentially yield similar psychometric functions in Figure S1. If that is indeed the case, it wouldn't be surprising if there are no multiplicative operations at the neuronal level.

The behavioral model using only the sensory evidence but ignoring prior values was corresponded to the standard psychometric function without block-dependent choice threshold (**Figure 2c and S1d**). The behavioral model using only prior values corresponded to our previous RL model without action value (**Figure S1g**). Our multiplicative RL model was better fit to mouse behavior than the psychometric function and the RL model without action value, suggesting that mice used the action values for making choices.

Although the behavioral analyses suggest the use of action values, our study found that the mPFC neurons additively represented the prior values and choices (**Page 8, line 259**).

While I acknowledge that this issue does not impact the authors' main conclusion regarding the global and local representations of priors, I would suggest that the term "integration" be handled with caution. Considering terms like "additive combination" might be more appropriate, but I'm open to seeing evidence that supports the use of "integration" if the authors disagree with this suggestion.

In summary, although the behavior data suggested the use of action value for value updating, we did not find clear evidence that the mPFC neurons represented action values. We found that the mPFC neurons additively represented the prior values and choices during sounds.

Thank you for the supportive comment. We decided not to use “integration” in the manuscript. Instead, we used “additive combination” of prior values and choices for the mPFC representation (**Page 8, line 259**).

We removed the sentences that the mPFC neurons represented the action values (**Title and Abstract; Page 3, line 78; Page 5, line 166; Page 13, line 434 and 458; Page 14, line 475; Page 15 line 525; Figure S15**).

Minor)

The authors added a logistic regression analysis that illustrates the effects of sounds and choice-outcome associations in past trials (Fig. S1c). It shows that the absolute coefficients for the sounds presented in past trials appear to be even larger than those for choice-outcomes.

1) It doesn't make sense to me that the sign of the coefficients becomes significantly reversed (negative) for the sounds in past trials. This suggests that animals tend to choose the left side when the previous sound was associated with a rightward choice in a tone cloud? If that is the case, is there any adaptive reasons that they take such a strategy in this particular task? Could you confirm if my understanding is correct?

Thank you for pointing out the figure. We re-checked the analysis and found that there was a problem. Because the sound frequencies in previous trials were highly correlated to the rewarded side, the regression coefficients for sounds were potentially contaminated with the coefficients for choices and rewards.

Based on a previous study investigating the effect of reward history on the current choice (Sugrue et al, science, 2004), we re-analyzed how the reward amounts of left and right choice in past trials affected the choice in our study (**Methods, Page 25 line 864**). We found that at least the previous 2 trials were significantly affected the choice (**Figure S1c**).

2) Does this indicate that the previous sounds have a stronger influence on choices than, or at least an influence almost equal to, past rewarded choices? Is it possible that the prior representations associated with the past reward is contaminated with the stimulus (or associated its choice side) history? Does this affect on their discovery of the past reward representations in AC?

As described in the answer to your minor comment 1, we apologize that our previous analysis potentially had some contaminations in the regression coefficients. In our current analysis, we found that the reward history of at least the previous 2 trials were significantly affected the choice in current trial (**Figure S1c**).

We also confirmed with the decoding analysis of area under the receiver operating curve (auROC) that the activity of auditory cortical neurons before sound was modulated by the prior value rather than the tone category (low or high) in the previous trial (below figure).

In the auditory cortical neurons, the decoding accuracy of prior value was better than that of tone category of previous trial before sound (Wilcoxon signed rank test). We evaluated the decoding performance with the area under the receiver operating curve (auROC) ranging between 0.5 and 1. The neurons are same as **Figure S5b** before sound.

Reviewer #2:

The revised paper has improved on the original and largely addressed my previous comments. The following points could benefit from further clarification.

We thank the reviewer for the supportive and helpful comments. We are sorry and appreciated that you point out some typos in our manuscript.

1. It remains unclear whether there is a “sequence” of activity after sound onset, and certain aspects of Supplementary Figure 3d are unclear. Since this is a minor point in the scope of the paper, it might be worth clarifying the meaning or removing emphasis on the sequence. Specifically:

As you advised, we removed the word “sequence” from the manuscript (**Abstract: Page 1, line 24; Page 5, line 166; Page 6, line 195; Page 15, line 526; Page 17, line 588; Figure S3 title and caption of Figure S3d (page 4)**).

This may be an issue of definitions: “sequence” can imply that there are middle-preferring neurons (e.g. neuron A, then B, then C). However, the heatmaps and shaded regions in Supplementary Figure 3d (right) look more like there are two groups: one which is active shortly after sound onset, and another which ramps up near the end of the sound. If this is indeed what the authors mean by “sequence” based on the context in lines 193-198, it would be helpful to explicitly state this. If not, it remains unclear whether there are middle steps in the sequence, since the spike shuffling method would only test against randomly distributed activity.

We removed the word “sequence” in previous lines 193-198. As you pointed, we clarified that there were at least two types of neurons. One group increased the activity around the sound onset, while the other group gradually ramped up the activity during sound (**Page 6, line 195**).

Which neurons are plotted in Supplementary Figure 3d? The text and legends seem to suggest that it is all task-relevant neurons as in Figure 3d, but the supplementary figure appears to be showing a subset of task-relevant neurons. Please clarify which neurons are being plotted, and how they are defined if it is a subset.

We analyzed a subset of task-relevant neurons which increased the activity between -0.1 and 1.0 s from sound onset in the long sound correct trials. Here we targeted the neurons activated before and during sound presentations. Same neurons were targeted in **Figure 3f, 4f, and Supplementary Figure 5c, 7, and 9f**.

In the caption of **Supplementary Figure 3d**, we added a sentence to clarify which neurons were analyzed in the figure (**Supplements, Page 4 line 67**).

How is the data in Supplementary Figure 3d normalized? It doesn't appear to be range-normalized as in Figure 3b. Please clarify this in the legend.

We re-checked the normalization of **Supplementary Figure 3d** and replaced the figure. Both in **Figure 3b** and **Supplementary Figure 3d**, we first applied a Gaussian filter with the standard deviation of 100 ms to the spike activity. We then averaged the filtered activity across trials and standardized between 0 and 1.

We added sentences to clarify how we normalized the spike activity (**Supplements, Page 4, line 68**).

The red lines in Supplementary Figure 3d make it difficult to see whether there are underlying sequences, ideally they would be removed.

Thank you for the comment. We removed the red line in **Supplementary Figure 3d**.

It is unclear how correlation was done between the split groups of trials: was the activity of all cells concatenated, or were correlations made separately for each cell and then combined?

The correlations in the figure were analyzed by randomly splitting all the long-sound trials into half. Same as above, we used the subset of task-relevant neurons activated between -0.1 and 1.0 s from the sound onset and analyzed as follows:

- 1: In each neuron, we averaged the activity across the half of trials and analyzed the maximum activity timing.
- 2: In the other half of trials, we also analyzed the maximum activity timings of the subset of task-relevant neurons.
- 3: Spearman correlation was analyzed between the maximum activity timings of neurons in the procedure 1 and 2.
- 4: The procedure 1–3 was repeated 100 times to reduce the noise from random grouping of trials. We averaged the Spearman correlations of 100 times to analyze the average correlation shown in the figure.

The randomization of spike activity (bottom figures) was done by shuffling the spike timings. We then split the shuffled spike activity into half and analyzed the correlations with the procedure 1–4.

We added the sentences to clarify how we analyzed the correlations in **Supplementary Figure 3d (Supplements, Page 5, line 73)**.

2. The new plots of activity by block in Figure 3f, 4f, and S8f are helpful, though it appears that the plots of activity by tone have dramatically changed from the original paper (e.g. Figure 3f (left) in the revised paper, compared to Figure 3e in the original paper). It is unclear what would have caused this change from the text and legends, could the authors clarify?

In the **previous Figure 3e**, the activity was plotted based on the neurons activated in each time window. Therefore, the neurons used in each time window were mostly same but different. On the other hand, the **current Figure 3f** had the same neurons in the entire time windows. The neurons in the current **Figure 3f** matched to the neurons in **Supplementary**

Figure 3d, 5c and 7. We modified the figure caption to clarify the current **Figure 3f (Page 17, line 611).**

3. Typos:

Figure 3f legend: “starndard”

Thank you so much for finding the typos. We revised as “standard” (**Page 17, line 611**).

Line 236: “ANOVA p value” – presumably this meant to include a p-value? Or if it is meant just to identify the test, it could read “x-way ANOVA”

We revised as “two-way ANOVA” (**Page 7 line 236; Page 17 line 615; Page 18 line 617 and 645**) (**Supplements, Page 8, line 116; Page 11 line 153**).

Reviewer #3:

The authors have done a very nice job with this revision, and have comprehensively addressed my comments with several additional analyses and textual clarifications. The additional paragraphs in the discussion are helpful and provide important context for interpreting the unique contributions of this study. My only additional comment is just to ask the authors to clean up the discussion paragraphs a bit, as it appears there are quite a few typos there.

With these small edits, I am happy to recommend the manuscript for publication, and would like to thank the authors for their commendable effort in carrying out these studies and revisions.

We thank the reviewer for the helpful comments. We appreciate the comments which improved and clarified our paper. Based on the comments, we revised typos and sentences in Discussion (**Page 13, line 439; Page 14 line 474, 481, 485**).

Reviewer #4:

I am overall happy with the study and the extensive analysis. However, a few changes is still necessary to me.

We thank the reviewer for carefully reading our paper and provide us the helpful comments.

1) *The authors changed terms related to “confidence-value” to “action value”. First of all, I really appreciate removal of “confidence-value” terms. As one the other reviewers also noted that term was confusing. The new term, however, is still a little bit confusing (with “choice”). “Expected value” or “Expected value of action” as in reference [13] are better choices to me. Alternatively, the authors could clarify the meaning of “action value” from the beginning (and also more explicitly). Currently, “action-value” is defined relatively early in the paper (line 75), but I think the term “action-value” has been used even before line 75.*

We agree your advice. The word “action value” is a terminology in a reinforcement learning (Sutton and Barto, 2015) but not a general word in the field of neuroscience. We added the definition of action value at the beginning of Introduction (**Page 2, line 37**).

2) *I am still not convinced that mPFC decodes expected value/action value as opposed to choice. The authors has acknowledged this in a few places in the paper, but looks like the main narrative is still about the mPFC mainly decodes the expected value while M2 mainly decodes choices. The fact that expected value has been observed in all areas later in the trial should be clearly stated from the beginning, and small differences between areas and speculations should follow after that. Notably, given the strong correlation between choice and expected value in this study, I do not even expect to see significant difference in decodability in any region.*

As you advised, we removed the claim that the mPFC represented action values (**Title and Abstract; Page 3, line 78; Page 5, line 166; Page 13, line 434, 458; Page 14, line 475; Page 15, line 525; Figure S15**).

We did not find clear evidence that the mPFC neurons “integrated” the sensory evidence and prior values for action values, but we only found that the neurons additively represented both the prior values and choices (**Figure S8**). As advised by another reviewer, the representation of action value was different from the representation of prior value + choice.

For the analyses of each brain area (**Figure 3, S9, and 4** for the mPFC, M2, and AC), we found that the neurons represented the prior values (**mPFC: page 8, line 259; M2: page 8, line 263; AC: page 9, line 299**), while the differences among three brain areas were analyzed in later figures (**Figure 5 and 6**).

3) *Most importantly, I do not think analysis that produces Table S1, and panel j of figures 3, 4, and S8 is correct at all. There is no statistical test in that analysis, and as the numbers show there is a huge variance in results. Moreover, “choice” is highly correlated with “prior + sound” so they can not be compared as features of a regression-based analysis. This*

analysis should be removed. Interestingly, results of this analysis (Table S1) also contradict with claims of the paper that M2 is mainly about choice, and AC is mainly about the sound.
Thank you for the advice. We removed the analysis.

We added a different analysis in **Figure S8b (Methods, page 28, line 975)**. To test the neural encoding of task variables including action values, a similar regression analysis was used in previous research (Samejima et al, Science 2005; Ito and Doya, J Neurosci, 2009; Sul et al, Nat Neurosci, 2011; Massi et al, neuron, 2018).

In the M2 and mPFC, we tested whether the activity of neurons in each time window represented the choice, prior, or choice + prior value. This analysis tested whether the neurons represented choices or additively represented choices + prior values. We then found the additive representation of choice and prior value in the mPFC.

Although “choice + prior value” and “action value” were highly correlated as you pointed, our regression analysis found that there was no evidence that the mPFC neurons represented the action values (**Figure S8b**) (**Page 3, line 80; Page 13, line 434**).

REVIEWERS' COMMENTS

Reviewer #1 (Remarks to the Author):

The authors have addressed my concerns. I recommend the paper for publication.

Reviewer #1 (Remarks on code availability):

The following link for their dataset in the figshare seems not available.

3. How to use

You can get access to the spike and behavioral data from the following cites:

DOI: 10.6084/m9.figshare.24319705

For the MATLAB code here to work, the above data are needed to convert into the appropriate intermediate files.

Reviewer #2 (Remarks to the Author):

The revised paper has addressed my previous comments and in my opinion is ready for publication.

Reviewer #4 (Remarks to the Author):

I wish to thank the authors for their response and edits. I do not have any major concern around this work anymore.

Reviewer #4 (Remarks on code availability):

The code seems to be clear, but it is mainly about visualization, and not useful without the data (I understand that the authors might not want to publish their data set online.)

We appreciate the reviewers' thoughtful and helpful comments. Our responses to their comments are provided below.

Reviewer #1:

The authors have addressed my concerns. I recommend the paper for publication.

We thank the reviewer for all the support and comments for improving our manuscript.

Reviewer #1 (Remarks on code availability):

The following link for their dataset in the figshare seems not available.

3. How to use

You can get access to the spike and behavioral data from the following cites:

DOI: [10.6084/m9.figshare.24319705](https://doi.org/10.6084/m9.figshare.24319705)

For the MATLAB code here to work, the above data are needed to convert into the appropriate intermediate files.

The DOI for datasets is now publicly available

[<https://doi.org/10.6084/m9.figshare.24319705>].

The codes are available on Github [https://github.com/funamizu-lab/Ishizu_et_al_2023]. We also uploaded the codes and the intermediate files on Zenodo to generate the figures in the manuscript [<https://doi.org/10.5281/zenodo.10881334>].

Reviewer #2:

The revised paper has addressed my previous comments and in my opinion is ready for publication.

We thank the reviewer for the supportive and helpful comments to improve our manuscript.

Reviewer #4:

I wish to thank the authors for their response and edits. I do not have any major concern around this work anymore.

We thank the reviewer for the supportive and helpful comments for improving our manuscript.

Reviewer #4 (Remarks on code availability):

The code seems to be clear, but it is mainly about visualization, and not useful without the data (I understand that the authors might not want to publish their data set online.)

The spike and behavioral data are available on figshare:

[<https://doi.org/10.6084/m9.figshare.24319705>].

The codes are available on Github [https://github.com/funamizu-lab/Ishizu_et_al_2023]. We also uploaded the codes and the intermediate files on Zenodo to generate the figures in the manuscript [<https://doi.org/10.5281/zenodo.10881334>].